

# Inverse modelling of New Zealand's carbon dioxide balance estimates a larger than expected carbon sink

Beata Bukosa[1], Sara Mikaloff-Fletcher[1], Gordon Brailsford[1], Dan Smale[2], Elizabeth D. Keller[3,4], W. Troy Baisden[5], Miko U.F. Kirschbaum[6], Donna L. Giltrap[6], Lìyǐn Liáng[6], Stuart Moore[1], Rowena Moss[1], Sylvia Nichol[1], Jocelyn Turnbull[3], Alex Geddes[2], Daemon Kennett[1], Dora Hidy[7], Zoltán Barcza[8], Louis A. Schipper[9], Aaron M. Wall[9], Shin-Ichiro Nakaoka[10], Hitoshi Mukai[10], Andrea Brandon[11]

[1]National Institute of Water and Atmospheric Research, Wellington, New Zealand

[2]National Institute of Water and Atmospheric Research, Lauder, New Zealand

[3]GNS Science, National Isotope Centre, Lower Hutt, New Zealand

[4]Antarctic Research Centre, Victoria University of Wellington, Wellington, New Zealand

[5]Te Pūnaha Matatini Centre of Research Excellence and Motu Research, Wellington, New Zealand

[6]Manaaki Whenua - Landcare Research, Palmerston North, New Zealand

[7]Excellence Center, Faculty of Science, ELTE Eötvös Loránd University, Martonvásár, Hungary

[8]Department of Meteorology, ELTE Eötvös Loránd University, Budapest, Hungary

[9]School of Science and Environmental Research Institute, University of Waikato, Private Bag 3105, Hamilton, 3240, New Zealand

[10]National Institute for Environmental Studies, Tsukuba, Ibaraki, Japan

[11]New Zealand Ministry for the Environment, Wellington, New Zealand

*Correspondence to*: Beata Bukosa (Beata.Bukosa@niwa.co.nz)

**Abstract.** Accurate national scale greenhouse gas source and sink estimates are essential to track climate mitigation efforts. Inverse models can complement inventory-based approaches for emissions reporting by providing independent estimates underpinned by atmospheric measurements, yet few nations have developed this capability for carbon dioxide ($CO_2$). We present results from a decade-long (2011-2020) national inverse modelling study for New Zealand, which suggests a persistent carbon sink in New Zealand's terrestrial biosphere ($-171 \pm 29$ Tg $CO_2$ yr$^{-1}$). This sink is larger than expected from either New Zealand's Greenhouse Gas Inventory ($-24$ Tg $CO_2$ yr$^{-1}$) or prior terrestrial biosphere model estimates ($-118 \pm 22$ Tg $CO_2$ yr$^{-1}$, Biome-BGCMuSo and CenW). The largest differences are in New Zealand's South Island, in regions dominated by mature indigenous forests, generally considered to be near equilibrium, and certain grazed pasture regions.



Relative to prior estimates, the inversion points to a reduced net $CO_2$ flux to the atmosphere during the autumn/winter period. The overall findings of this study are robust with respect to extensive tests to assess the potential biases in the inverse model due to transport error, prior biosphere, ocean and fossil-fuel estimates, background $CO_2$ and diurnal cycles. We have identified $CO_2$ exchange processes that could contribute to the gap between the inverse, prior and inventory estimates, but the magnitude of the fluxes from these processes cannot entirely explain the differences. Further work to identify the cause for

the gap is essential to understand the implications of this finding for New Zealand's inventory and climate mitigation strategies.

## 1 Introduction

Under the Paris Agreement, each nation is required to set, track and report progress against a nationally determined contribution towards meeting the goals of the agreement. Where nations set emissions related targets, they must follow

agreed reporting guidelines (UNFCCC, 2018) and adhere to inventory methodologies set out by the Intergovernmental Panel on Climate Change (IPCC) (IPCC, 2006, 2019). Current greenhouse gas emissions reduction strategies rely on tracking greenhouse gas emissions and carbon uptake from local to global scalMOLes, with a special focus on national scale actions to limit emissions and, in doing so, limit increases in global average temperatures (UNFCCC, 2015). National greenhouse gas inventory reporting is typically based on bottom-up nationally representative data collection methods. The most recent

IPCC guidelines refinement (IPCC, 2019) recommends using independent methods, such as atmospheric inverse models (i.e., top-down methods) as a complementary tool to estimate national scale carbon fluxes. Top-down methods rely on atmospheric measurements and relate atmospheric observations to fluxes from specific regions through atmospheric transport model simulations.

National scale inverse modelling has been successfully used for methane and other greenhouse gases in a number of

countries (Manning et al., 2011; Miller et al., 2013; Ganesan et al., 2015; Henne et al., 2016; Maasakkers et al., 2021; Lu et al., 2022). Yet, to date, only three countries have successfully used this approach to estimate national scale carbon dioxide ($CO_2$) fluxes enabling comparisons with estimates reported in national greenhouse gas inventories: New Zealand (Steinkamp et al., 2017), United Kingdom (White et al., 2019) and Australia (Villalobos et al., 2023). While $CO_2$ inverse modelling studies have supported $CO_2$ flux estimation on larger scales (Gerbig et al., 2003; Matross et al., 2006; Schuh et al., 2010;

Meesters et al., 2012; Deng et al., 2021; Byrne et al., 2023; Kou et al., 2023), their application on a national scale is still subject to limitations (Byrne et al., 2023). Top-down national scale $CO_2$ estimates are impacted by limited data coverage, transport model errors, inaccurate representation of the diurnal cycle, uncertainties in $CO_2$ background values, as well as other factors, all of which contributes to the complexity of interpreting top-down $CO_2$ methods for improving carbon flux estimates. This complexity is further compounded by the fact that national greenhouse gas inventories are restricted to

anthropogenic emissions, making it even more challenging to accurately compare the two methods.



New Zealand's unique geographical advantages (i.e., isolated landmass, far from other terrestrial sources or sinks) and current technical capabilities (i.e., high resolution modelling, high precision $CO_2$ observing sites) make it an excellent case study to develop, test and adapt national scale top-down methodologies. Steinkamp et al. (2017) developed the first top-down national scale inverse model for $CO_2$ focusing on New Zealand's carbon budget between 2011 and 2013. They estimated a national sink of -98 ± 37 Tg $CO_2$ yr$^{-1}$ from the terrestrial biosphere, which is larger than that reported in New Zealand's Greenhouse Gas Inventory (1990-2022) (the 2024 Inventory, MfE (2024)) and prior bottom-up estimates. In particular, the inversion suggested that the south-west of New Zealand (referred to as the Fiordland region) was a large $CO_2$ sink. Fiordland is a region dominated by mature, indigenous forests, which are traditionally assumed to be carbon neutral (Kira & Shidei, 1967; Odum, 1969; Luyssaert et al., 2008; Holdaway et al., 2017; Paul et al., 2021). The study suggested that these environments might have a much greater potential to absorb carbon than thought previously. The authors recognised that top-down and bottom-up inventory methods are not directly comparable due to scope and methodological differences and sought to resolve discrepancies. However, after adjustments had been made for these differences between methods, the top-down results still pointed to a larger sink than inventory methods (Steinkamp et al., 2017).

The stronger than expected New Zealand $CO_2$ sink reported in Steinkamp et al. (2017) was intriguing. However, there were three key limitations to that study. The study only covered three years (2011-2013), which made it impossible to assess whether the sink was persistently larger than that reported in New Zealand's Greenhouse Gas Inventory or the result of variability. In addition, the study was based on a single model at ≈12 km spatial resolution, which may not be able to adequately represent the airflow in parts of New Zealand due to its complex topography (Landcare Research, 2010b). The inversion also relied on only one prior biospheric model (Biome-BGC) that was not calibrated for New Zealand's specific forest biomes.

Here, we present the results of a decade long (2011-2020) national scale $CO_2$ inverse model. We improved the transport model resolution used in Steinkamp et al. (2017) by a factor of ten and tested the sensitivity of our results to the choice of model. In addition, we used two prior terrestrial biosphere models, which have been tuned using country specific data. We discuss the temporal and spatial change of the $CO_2$ fluxes, explore the underlying processes leading to the resulting sink and highlight differences between our top-down results and bottom-up estimates from both the biosphere models and 2024 Inventory. We performed a series of sensitivity tests to ensure the robustness of our results. Our work highlights the importance of the continuous advancement and the contribution that top-down methods provide when improving how we estimate and report national scale $CO_2$ fluxes, providing independent assurance of the environmental and scientific integrity of our climate mitigation efforts.



## 2. Methods

We used atmospheric surface $CO_2$ measurements (Sect. 2.1) to estimate net air-sea and air-land $CO_2$ exchanges with a Bayesian approach based on an inversion system described in Steinkamp et al. (2017) and detailed in Sect. S1. Here, we provide a brief overview of the system and focus in greater detail on key advances undertaken for this work, namely a major
advance to the atmospheric transport modelling and a priori estimates, particularly the terrestrial biosphere model.

In brief, the fluxes were estimated for 25 geographic regions (Fig. S1) on a weekly scale. We used a Lagrangian dispersion transport model (Sect. 2.2) to simulate the pathway of the air before it reached the measurement sites. We used prior terrestrial, oceanic and anthropogenic fluxes (Sect. 2.3) and their uncertainty estimates (Sect. 2.4) to optimise the fluxes.

**2.1 Sites, measurements and background**

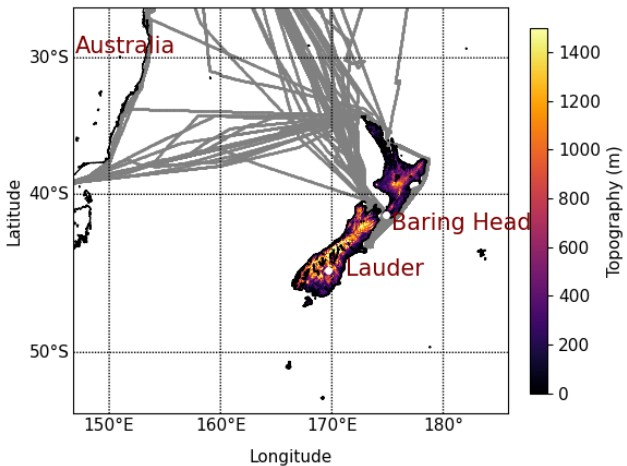

**Figure 1. New Zealand's topography (zoomed in version Fig. S5) with location of New Zealand long running *in situ* measurement sites, Baring Head and Lauder. Ship based measurements collected onboard the Trans Future 5 ship were used to help characterise the $CO_2$ in the background air (grey lines).**

The posterior fluxes in the inversion were estimated from atmospheric surface measurements at two sites in New Zealand, Baring Head (Fig. 1 and Fig. 2, North Island, 41.408∘ S 174.871∘ E) and Lauder (South Island, 45.038∘ S, 169.684∘ E) (Lowe et al., 1979; Stephens et al., 2011; Brailsford et al., 2012; Stephens et al., 2013; Steinkamp et al., 2017; Smale et al., 2019). Baring Head is a coastal site, fully exposed to winds from the South and near the relatively narrow Cook Strait, while Lauder is an elevated inland location in the lee of the Southern Alps on an expansive plain (Brailsford et al., 2012;
Steinkamp et al., 2017). We used hourly mean measurements averaged over 13:00 to 14:00 and 15:00 to 16:00 local time



(Fig. 2a and 2b) when the air is well-mixed so that the $CO_2$ signal is representative of regional processes. The characteristics of the Baring Head and Lauder $CO_2$ instruments and meteorological conditions are described in detail in Sect. S2 and 2.2.1.

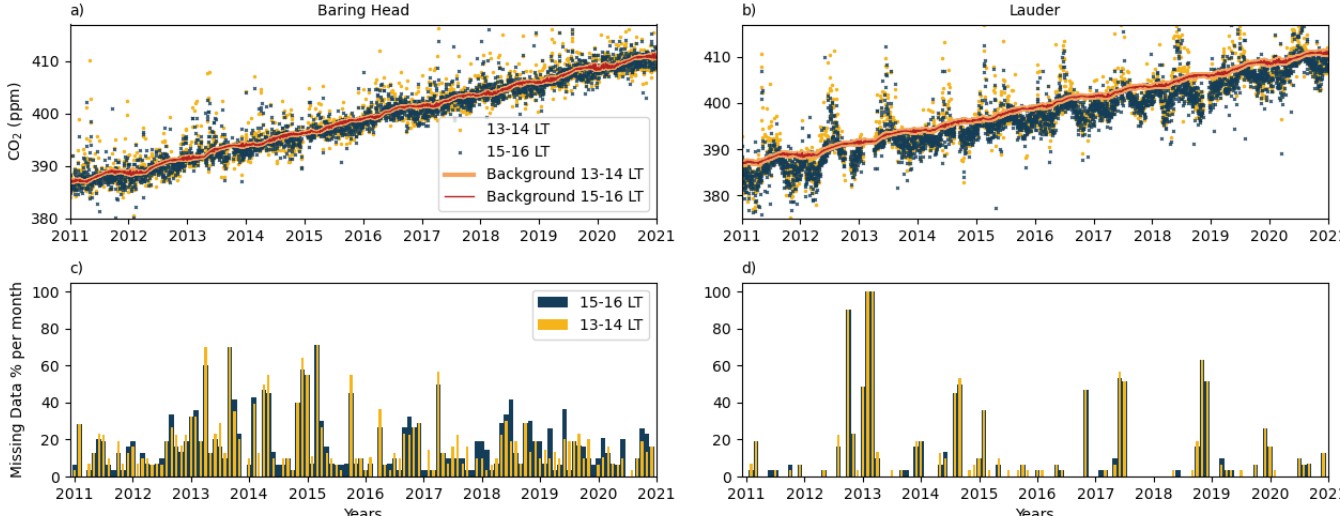

**Figure 2. $CO_2$ measurements at Baring Head (a) and Lauder (b), yellow and blue dots represent measurements at 13:00-14:00 and 15:00-16:00 local time, the orange and red line represents the weighted southern-northern background based on Baring Head and TF5 measurements (13:00-14:00 and 15:00-16:00 local time). Data gaps are shown in c) at Baring Head and d) at Lauder.**

New Zealand is located far away from other land masses and surrounded by approximately 2000 km of ocean in all directions which simplifies the construction of accurate $CO_2$ background. Background $CO_2$ represent the $CO_2$ mole fractions
reaching New Zealand before perturbation by local influences while any deviations from those background $CO_2$ mole fractions are representative of processes occurring in New Zealand. We constructed the background values based on measurements collected at Baring Head to represent southerly background conditions (Manning & Pohl, 1986) and measurements on board the Trans Future 5 ship (TF5, cruising between Japan-Australia-New Zealand, held by National Institute for Environmental Studies (NIES) as the Volunteer Observing Ship (VOS) program (Terao et al., 2011; Yamagishi
et al., 2012; Müller et al., 2021)) to represent northerly background conditions (Fig. 1). The data time series used for the inversion was constructed by subtracting background measurements from the afternoon measurements at the two sites (Fig. S2).

## 2.2 Atmospheric transport model

A Lagrangian dispersion model, NAME III (Numerical Atmospheric dispersion Modelling Environment, Jones et al.
(2007)), was used to back-calculate the pathway of the air before it arrived at Baring Head and Lauder between 13:00-14:00



and 15:00-16:00 local time. We used the atmospheric transport model to link the regional total fluxes with the data time series (Steinkamp et al., 2017).

NAME III is driven by meteorological inputs from the National Institute of Water and Atmospheric Research (NIWA) operational numerical weather prediction (NWP) models. These are the New Zealand Limited Area Model
(NZLAM, ≈12 km spatial resolution) covering the 2011-2013 inversion period (Steinkamp et al., 2017) and the New Zealand Convective Scale Model (NZCSM, ≈1.5 km spatial resolution) used for the period mid 2016-2020 (Webster et al., 2008). Both models are specific configurations of the UK Met Office Unified Model (UM) (Davies et al., 2005). To fill a gap in the archived data, a custom NZCSM was configured and run in hindcast mode to generate the required NAME III input data for the period 2014-mid 2016 (hereinafter referred to as NZCSM-like, Sect. S3).

NZCSM provides approximately ten times higher horizontal spatial resolution compared with the previous work using NZLAM, which allowed us to more accurately resolve the air flows over and around New Zealand's complex terrain, especially around the South Island Southern Alps (Fig. S5). Furthermore, at the resolution used by NZCSM, we started to be able to resolve some of the convective processes. This made it possible to run the model without a convection parameterisation scheme and, instead allowed the model dynamics to explicitly deal with convective initiation. The
importance of model resolution in inversion methods has been highlighted previously (Bergamaschi et al., 2005; Baker et al., 2006; Prather et al., 2008), so the mid-term switch from NZLAM to NZCSM-generated input data was considered worthwhile to obtain the best possible wind climatology for this study.

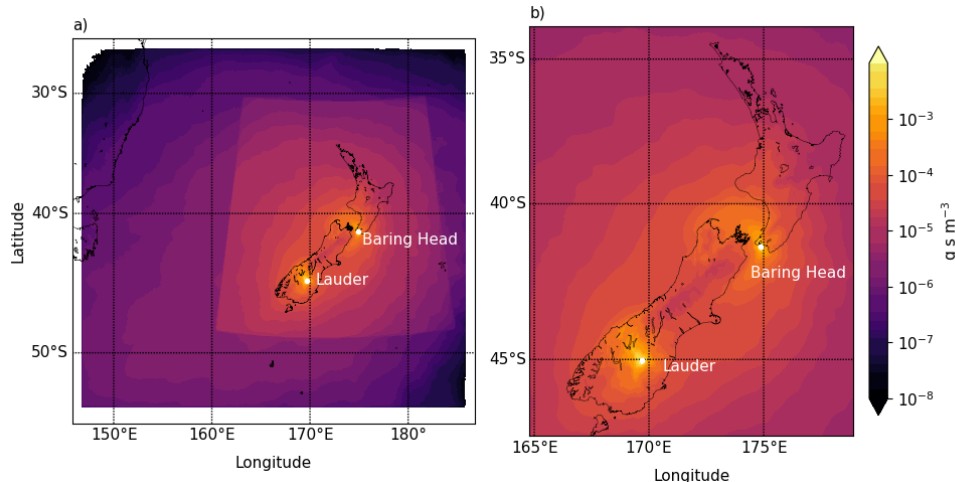

**Figure 3. Combined NAME III air concentration (i.e., footprints) based on Baring Head and Lauder for 2011-2020 at 13:00-14:00**
**and 15:00-16:00 local release time. 2011-2013 footprints are based on NZLAM while 2014-2020 is based on NZCSM meteorology input. NZCSM covers a smaller domain relative to NZLAM. The full domain is shown on plot a) and a zoomed version on plot b).**

Previous greenhouse gas inversion studies have used both time integrated (Manning et al., 2011; Steinkamp et al., 2017) and disaggregated modelled footprints (Gerbig et al., 2003; White et al., 2019). Our inversion used 4-day integrated





air concentration (i.e., footprints, units g s m⁻³, Fig. 3), averaged throughout the Planetary Boundary Layer (PBL). Forced by
the NWP data described above, we performed 4-day backward simulations with the NAME III model by releasing 10,000
$CO_2$ particles during a 1-hour period from Baring Head and Lauder each calendar day for 13:00-14:00 and 15:00-16:00 local
time. Based on the pathway of the particles (air concentration) the inversion linked each measurement point with the regions
and land types that influenced the measured $CO_2$ signal, resulting in higher ($CO_2$ source regions) or lower ($CO_2$ sink regions)
values. By the end of the 4 days, most of the particles had left the model domain (Steinkamp et al., 2017).

The modelled footprints highlight the sensitivity of the inversion to different parts of New Zealand. Based on the
average 2011-2020 footprints (Fig. 3 and S4), the two sites are sensitive to most of the South Island and southern part of the
North Island; however, our current network does not fully cover the northern parts of the North Island and central part of the
South Island. The inversion in the years 2014-2020 used the NZCSM model with a smaller footprint domain (Fig. 3a), hence
Australia and certain ocean regions were masked out. The earlier 2011-2013 inversions were based on the NZLAM model
with its larger footprint domain that included the east coast of Australia. Later in Sect. 4.1 we will show that the smaller
domain did not impact our inversion results.

### 2.2.1 Atmospheric transport model validation

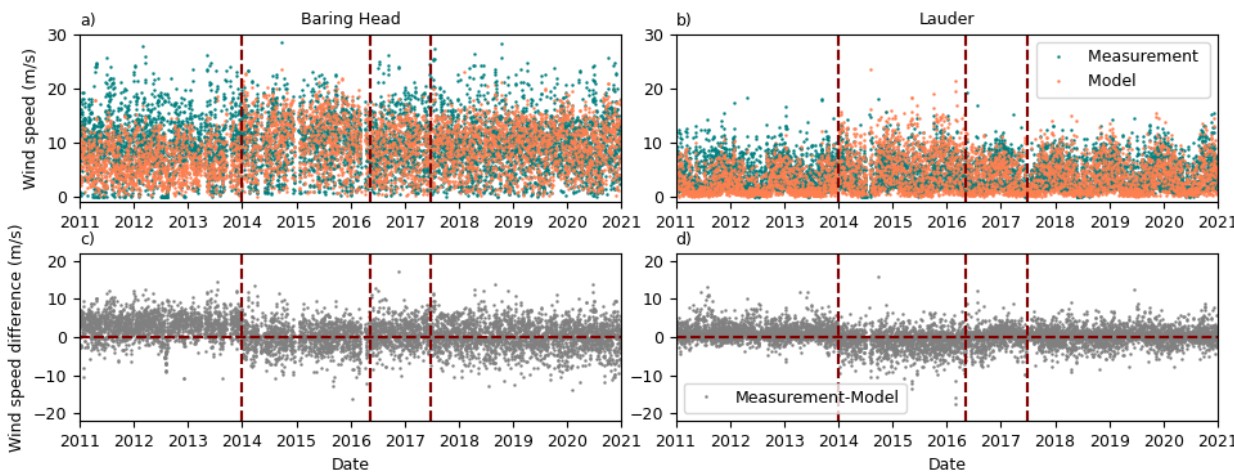

**Figure 4. Hourly measured and modelled wind speed at Baring Head (a) and Lauder (b) and their difference (c and d,**
**measurement-model) at 13:00-14:00 and 15:00-16:00 local time (as on Figure 2). The vertical lines highlight the change between**
**different models or model version (NZLAM: 2011-2014, NZMCS-like: 2014- mid 2016, NZCSM "pre-ENDGAME": mid 2016-mid**
**2017 and NZCSM "ENDGame": mid 2017-2020).**

We used four different models (NZLAM, NZCSM-like, NZCSM "pre-ENDGame" and NZCSM "ENDGame". Sect. 2.2. and
S3) as meteorological input in the NAME III atmospheric transport model. The change between different models was driven



by model improvements and updates. Here, we analyse the performance of the meteorological input data by comparing the modelled data with measured meteorological variables at the two $CO_2$ measurement sites. Note, the geographical characteristics of the two sites are quite different (Sect. 2.1, Steinkamp et al. (2017)).

NZLAM (used until 2014) consistently underestimated the wind speed at both sites (Fig. 4). The performance of both the NZCSM-like and operational NZCSM model led to better agreement with the measured wind speed; except, the
NZCSM-like modelled wind speeds (used from 2014-2016) were overestimated at Lauder. The accuracy of the modelled wind conditions is a critical factor in the accurate estimation of $CO_2$ fluxes. Moreover, precise modelled wind directions are crucial to accurately link the measured $CO_2$ signal with the regions that impacted the atmospheric $CO_2$ levels. Figure 5 shows the measured and modelled wind speed and direction with both NZLAM and NZCSM for year 2018, when both model outputs were available. Other years follow a similar pattern (Fig. S7). NZLAM showed a consistent bias in the wind
direction for both sites, that could lead to the misattribution of $CO_2$ source and sink regions when NZLAM was used (years 2011-2013). As discussed in Sect. 2.2 this bias was impacted by the coarser model resolution of NZLAM. For Baring Head, NZLAM suggested a dominant wind direction from the Northwest (instead of North), while for Lauder NZLAM suggested a dominant wind direction from the North and Northwest (instead of Northeast). Updating the model to NZCSM significantly improved the modelled wind direction and speed, reducing the model uncertainties for the post-2013 inversion years.
Comparison of radiosonde PBL measurements at Lauder showed that all models underestimated the measured PBL (Sect. S3.1 and Fig. S6). In Sect. 4 we will further analyse the impact of the transport model on the estimated fluxes.

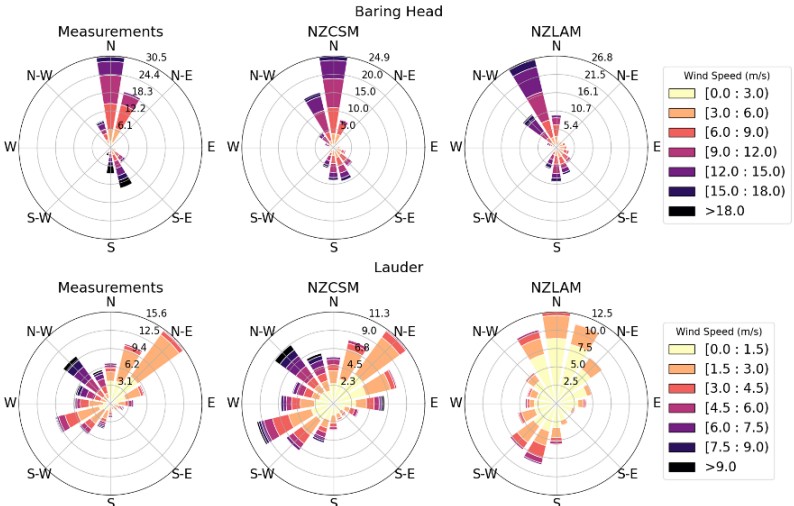

**Figure 5. 2018 mean measured and modelled (NZLAM and NZCSM) wind roses at Baring Head and Lauder when both modelled data was available.**



## 2.3 Prior information

We used prior oceanic, anthropogenic and biospheric fluxes (Table 1, Fig. 6 and Fig. 7) to estimate the weekly total posterior fluxes. $CO_2$ emissions from other processes, such as biomass burning and $CO_2$ chemical production (Bukosa et al., 2023) at the surface were presumed to be minor in New Zealand, and we excluded them from our inversion system.

### 2.3.1 Prior oceanic and anthropogenic fluxes

Oceanic fluxes (Fig. 6a and 6c) were obtained from Landschützer et al. (2020a); Landschützer et al. (2020b) based on monthly open-ocean air-sea $CO_2$ fluxes. They were further merged with calculated coastal fluxes based on the climatology of surface coastal $pCO_2$. The open-ocean fluxes were compiled at a resolution of 1° x1°, while coastal $pCO_2$ values were available at a finer 0.25° x 0.25° resolution. Open-ocean flux estimates were only available up to 2019. The calculations of the 2020 open-ocean and 2011-2020 coastal $CO_2$ fluxes are described in Sect. S4.

We used annual mean fossil fuel $CO_2$ emissions from the Emission Database for Global Atmospheric Research (EDGAR) v7.0 (Crippa et al., 2022). EDGAR data were available on a 0.1° x 0.1° horizontal resolution with year specific emissions for the whole inversion period. The annual EDGAR emissions over mainland New Zealand were additionally scaled to the annual gross emissions (energy, industrial process and product use, agriculture, waste) reported in the Inventory (MfE, 2023) (Fig. 6d). Note, we did not optimise fossil fuel emissions as the estimated $CO_2$ signal from these emissions was subtracted from the measurements at the two sites beforehand.

**Table 1. Priors used for the inversion with base years, temporal and spatial resolutions.**

| Prior | Base Year | Native Temporal resolution | Native Spatial resolution |
|---|---|---|---|
| *Ocean* | | | |
| Open-ocean Landschützer et al. (2020a) | 2011-2019 | Monthly | 1° x 1° |
| Coastal-ocean Landschützer et al. (2020b) | 1998-2015* | Monthly | 0.25° x 0.25° |
| *Fossil Fuel* | | | |
| EDGAR v7.0** | 2011-2020 | Annual | 0.1° x 0.1° |
| *Biosphere* | | | |
| Biome-BGCMuSo v6.1 | 2011-2020 | Daily | 0.05° x 0.05° |
| CenW v6.0 | 2011-2020 | Daily | 0.05° x 0.05° |

*Climatology, **Scaled to 2023 Inventory



## 2.3.2 Prior biospheric fluxes

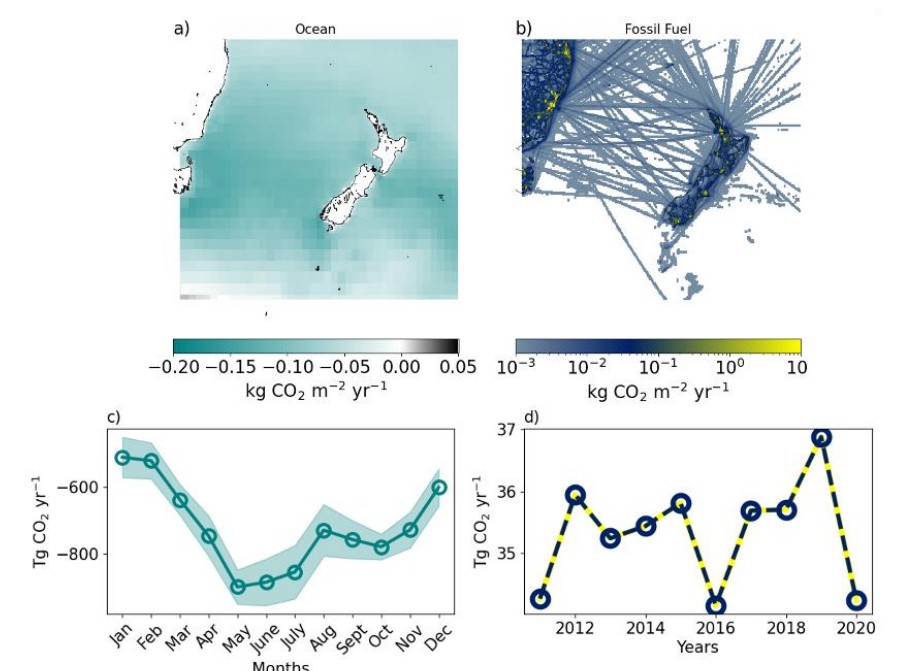

**Figure 6. 2011-2020 mean spatial distribution and time series of the CO₂ oceanic fluxes based on Landschützer et al. (2020a); Landschützer et al. (2020b) (a, c) and fossil fuel fluxes from EDGARv7.0 (b, d). The shaded areas on c) are 1 standard deviation of the 2011-2020 mean ocean fluxes. The fossil fuel flux time series (d) is based on annual emissions as reported in the 2023 Inventory that were used to scale the EDGAR emissions over mainland New Zealand. Note, negative numbers indicate CO₂ uptake.**

We have developed new prior biospheric (terrestrial) fluxes from two different models, the Biome-BGCMuSo v6.1 and CenW v6.0 (Carbon, Energy, Nutrients and Water), both optimised with country specific data.

Biome-BGCMuSo (http://nimbus.elte.hu/bbgc/) (Hidy et al., 2016; Hidy et al., 2022) is a biogeochemical terrestrial ecosystem model that simulates biological and physical processes controlling the carbon, nitrogen, and water cycles and fluxes between the atmosphere, plants and the soil. It is a branch of the Biome-BGC model developed by Numerical Terradynamic Simulation Group (NTSG) at the University of Montana (http://www.ntsg.umt.edu/project/biome-bgc.php) (Running & Coughlan, 1988; Thornton et al., 2002; Thornton & Rosenbloom, 2005; Thornton et al., 2005), which has been widely used to simulate the growth and carbon exchange of forests and grasslands in Europe and North America (Running & Coughlan, 1988; Running & Gower, 1991). Biome-BGCMuSo v6.1 represents a significant advance on previous versions of the model (Sect. 2.3.3). The model takes climate inputs of daily minimum and maximum air temperature, average daylight air temperature, precipitation, daylight vapour pressure deficit, and daylight solar radiation. Fore New Zealand implementations (Keller et al., 2014; Keller et al., 2021; Villalobos et al., 2023) these variables were obtained from NIWA's



New Zealand's Virtual Climate Station Network (VCSN), a 0.05◦ x 0.05◦ gridded data product that covers all of New Zealand from 1972-present (Tait et al., 2006; Cichota et al., 2008; Tait, 2008; Tait & Liley, 2009; Tait et al., 2012). Site specific soil information (texture, pH, and rooting depth) was obtained from the Fundamental Soil Layers database (Landcare Research, 2010a), which was re-gridded to match the VCSN. The Biome-BGCMuSo model provided fluxes for

five biomes across New Zealand: dairy pasture, sheep and beef pasture, ungrazed grassland, shrub, and evergreen broadleaf forest. Fluxes for the dairy and sheep and beef biomes were calibrated and validated for New Zealand based on eddy covariance (EC) data (Sect. S5 and Villalobos et al. (2023)). The remaining biome parameters were not optimized due to the lack of suitable EC data, and default parameters were used.

        The CenW v6.0 model (Kirschbaum, 1999; Kirschbaum & Watt, 2011) is a generic forest growth model that

provided $CO_2$ fluxes for radiata pine (*Pinus radiata*) that represents 90% of New Zealand's plantation forests (MfE, 2024). The CenW model also uses climate inputs from NIWA's VCSN. Like Biome-BGCMuSo, CenW also uses daily records of minimum and maximum air temperatures, precipitation, solar radiation and atmospheric humidity as well as other input data about water-holding capacity, soil texture and soil nitrogen concentration (Kirschbaum and Watt, 2011). Pine forests were parameterised in CenW based on 1309 individual observations from 101 sample plots situated across New Zealand

(Kirschbaum & Watt, 2011). These observations covered the growth of stands under various stand conditions, especially climatic conditions, and plantation ages.

        Both Biome-BGCMuSo and CenW produced daily estimates of Gross Primary Production (GPP), Ecosystem Respiration (ER) and Net Ecosystem Exchange (NEE = - (GPP - ER)) for each 0.05◦ pixel of their biomes covering the whole country for 2011-2020. We used a land-cover map to quantify the contribution by the modelled biomes to the

combined flux from each 0.05◦ pixel across the whole of New Zealand (Sect. S5, Table S1 and S2). Land-cover types were derived from the New Zealand Land Cover Database (LCDB) v5.0 (Landcare Research, 2020) and the LUCAS Land Use Map 2016 (MfE, 2016). The land cover categories were mapped into a final 10 category map (Fig. 8a, Steinkamp et al. (2017)) and matched with five Biome-BGCMuSo biomes (dairy pasture, sheep and beef pasture, ungrazed grassland, shrub and evergreen broadleaf forest) and *P. radiata* from the CenW model. The CenW *P. radiata* modelled by CenW was

assumed to be representative of New Zealand's plantation forests, the Biome-BGCMuSo evergreen broadleaf forest category was used to model the 'other forests' category (i.e., mostly indigenous forests) and we have used ungrazed grassland fluxes for the 'other grassland' category. We assumed zero flux for non-modelled artificial surfaces, bare and lightly vegetated surface, water bodies and croplands, which together represent only a small portion of the total land area (Fig. 8a). Table S3 shows the proportion of each category in all regions. The resulting spatial distribution, monthly and annual contribution of

the biomes is shown in Fig. 7.



### 2.3.3 Prior terrestrial model improvements

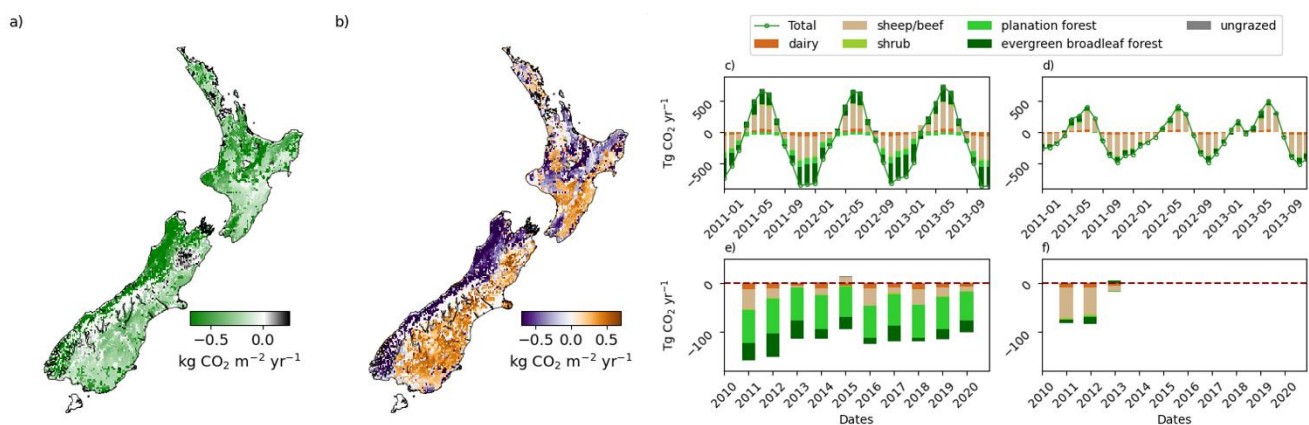

**Figure 7. 2011-2013 average spatial distribution of the prior bottom-up models: Biome-BGCMuSo merged with CenW fluxes (a), the difference between Biome-BGCMuSo merged with CenW fluxes and the original Biome-BGC model (b). The middle plots (c: Biome-BGCMuSo merged with CenW and d: Biome-BGC) show the monthly contribution of the fluxes for 2011-2013 when all three models were available, while the bottom plots (e, f) show the 2011-2020 annual contribution of the fluxes based on the Biome-BGCMuSo and CenW combined biomes (e) and Biome-BGC only (f). Note, data from Biome-BGC was available for 2011-2013 only (Steinkamp et al., 2017).**

For the previous inversion study of Steinkamp et al. (2017), focusing on years 2011-2013, all biomes were modelled with a single model, Biome-BGC v4.2 (Thornton et al., 2005; Keller et al., 2014). Here, we use additional fluxes that are representative of plantation forest fluxes from an independent model (CenW) that had been parameterised for *P. radiata* stands in New Zealand based on a comprehensive data set of available observations (Kirschbaum & Watt, 2011). Additionally, we use Biome-BGCMuSo fluxes representative of ungrazed grassland, that were not available in Biome-BGC. Biome-BGCMuSo improvements over the original Biome-BGC model include the explicit representation of management practices (such as harvesting, mowing, grazing, irrigation, etc), a 10-layer soil module (as opposed to just a single layer), more detailed nitrogen dynamics (including nitrification/denitrification processes), separate carbon and nitrogen pools for soft-stem plant tissue in addition to the existing pools for roots, leaves, leaf litter and woody stem, and implementation of plant drought stress and senescence (Hidy et al., 2016; Hidy et al., 2022). Soil hydrology has also been significantly improved (Hidy et al., 2022).

Updating Biome-BGC to Biome-BGCMuSo and CenW had a significant impact on the spatial distribution of the fluxes (Fig. 7a,b), leading to stronger $CO_2$ uptake in forested regions and weaker uptake in regions covered by sheep/beef grasslands. We found that the amplitude of the seasonal cycle in the original Biome-BGC model (Fig. 7d) was significantly smaller than the updated models, which has a strong impact on the seasonal cycle of the posterior fluxes in regions with low sensitivity to the measurement network. Using the CenW *P. radiata* fluxes instead of the Biome-BGCMuSo evergreen



needleleaf forest fluxes led to a stronger $CO_2$ uptake in all regions covered by plantation forests. Merging the CenW pine

forest fluxes with Biome-BGCMuSo led to a year-round net uptake by plantation forests (Fig. 7c, Fig S8) and stronger total

annual uptake (Fig. 7e). The impact of CenW is limited to a small part of the country covered by the plantation forest

category (Fig. 8a), albeit with very large per-unit consequences.

          For 2011-2013, the original Biome-BGC prior estimates used in Steinkamp et al. (2017) suggested a -59 ± 34 Tg

$CO_2$ yr$^{-1}$ national scale uptake. Using the Biome-BGCMuSo and CenW fluxes led to a -140 ± 22 Tg $CO_2$ yr$^{-1}$ net uptake,

more than double of the Biome-BGC estimates. The prior Biome-BGCMuSo and CenW fluxes also suggest an overall

stronger annual national sink than estimated by the inversion in Steinkamp et al. (2017) (-98 ± 37 Tg $CO_2$ yr$^{-1}$). However, in

Steinkamp et al. (2017), the additional $CO_2$ uptake was located in regions covered by mature forests, while in this study the

increased uptake in the prior flux estimates originate from regions covered by plantation forests. These model changes can

influence the posterior flux estimates, and we will further discuss their impact on the inversion results in Sect. 4.

## 2.4 Prior and posterior uncertainties

          The inversion used prior uncertainty estimates to weight and balance the information between the atmospheric

measurements and prior fluxes. We used the square root of the diagonal elements of the posterior covariance matrix (i.e.,

standard deviation, Sect. S1) as the posterior flux uncertainty for each model region. When aggregating the individual

regional posterior uncertainties into larger regions (i.e., North Island), we fixed (i.e., summed) 50% of the uncertainty term.

          The measurement uncertainty at Baring Head and Lauder was calculated as 1 standard deviation of the hourly

measurement interval that incorporates both atmospheric and measurement variability within the hour. The background

uncertainty estimates were based on the monthly standard deviations of the *in situ* data as well as differences between the

measurements and the seasonal time series decomposition by the Loess algorithm smoothed curve. In addition, these

uncertainties were weighted in the same way as the background data described in Sect. S2. The data and background

uncertainties were combined to give the total uncertainty applied to each data point as the root mean square (quadrature) of

the two uncertainties. For uncertainties in the transport model as well as possible errors in the fossil fuel emission estimates,

we assumed a minimum data uncertainty of 0.4 ppm. Lastly, we multiplied the final uncertainty by 3.9 based on the reduced

chi-squared statistic (fit of the inverse model to the observations, Gurney et al. (2004)). We populated the main diagonal of

the data covariance matrix with the square of the final uncertainty while off-diagonal elements of the data covariance matrix

were set to zero, hence we assumed no correlation between pairs of data points.

          We used the individual flux components (GPP and ER), instead of only NEE to define the prior terrestrial

uncertainties. This approach mitigates low uncertainties at times, especially in spring and autumn, when fluxes were very

small and could switch between negative and positive. It also provided a better representation of the $CO_2$ seasonal cycle in

the uncertainty term (i.e., leading to lower uncertainties in winter when both GPP and ER were small). The Biome-



BGCMuSo $CO_2$ flux uncertainties from dairy, sheep and beef and ungrazed grasslands were assumed to be 25% of their flux magnitude for NEE, GPP and ER. We assigned 30% for shrub and 50% to evergreen broadleaf forest. A higher uncertainty was assigned to these biomes because they were not parameterised with data from New Zealand. The uncertainty for pine fluxes from CenW were assumed to be 30%, 30% and 60% of the NEE, GPP and ER flux magnitudes, respectively. The

uncertainties from the individual GPP and ER components were then merged as the quadrature sum and scaled to the uncertainty magnitude of the NEE fluxes, to get an uncertainty of the net fluxes for each biome and mapped based on the land-cover map (Sect. 2.3). We have applied an additional 10% for each biome to account for uncertainties in the spatial pattern of the prior fluxes and land use map. The ocean prior uncertainty was estimated to be 50% of the flux magnitude for the coastal regions and 20% for all other regions (Roobaert et al., 2018). For the Australian, region we assumed zero prior

fluxes with a high uncertainty of 1000 Tg $CO_2$ $yr^{-1}$ as in Steinkamp et al. (2017). This meant that the posterior fluxes were entirely dependent on fluxes inferred from the inversion analysis without constraint by prior assumptions. We obtained the final prior uncertainties for each of the 25 regions by aggregating the grid-scale uncertainty estimates assuming full spatial correlation. The diagonal prior covariance matrix contains the regional uncertainty estimates and off-diagonal elements were set to zero.

**3 Regional CO₂ fluxes**

On average, both the prior and posterior North Island fluxes showed a similar net $CO_2$ sink (Fig. 8b,c,d, Fig. 9, Figs. S9-S11, Table S4-S7). The strongest $CO_2$ uptake originates from the central-eastern part of the North Island (region 4 and 5, Fig. 9), which is dominated by a mixture of planted exotic and indigenous forests (Fig. 8a). However, the northern parts of the country are only weakly constrained by the measurements network (Sect 4.4) and therefore fluxes from these regions

remain close to the prior estimates. The majority of exotic forests in New Zealand grow in this area, and the strong uptake in the exotic forest regions is largely driven by the prior CenW pine fluxes, which are characterised by a strong net annual $CO_2$ uptake.

In contrast to the North Island, there were large differences between the prior and posterior estimates for the South Island. The posterior fluxes suggested strong $CO_2$ uptake along the west coast of the South Island, especially in the southern

part (Fig. 8c). Large sinks were estimated in regions covered by forests, while other regions as the north-eastern (region 10, Canterbury) and the central-eastern region (region 12) did not show a strong sink activity. These regions are mostly dominated by grasslands. Although our prior fluxes suggest that they are weak sinks, our posterior fluxes point to a mixed carbon exchange scenario. Regions 14 (Lauder) and 15 (Otago and Southland region) are also dominated by grassland (mostly sheep and beef pasture) but showed a relatively large sink. ~10% of region 15 is also covered by a mixture of

plantation and mature forests. Note, region 14 was only designed as a region around the $CO_2$ measurement site to capture the local $CO_2$ processes.



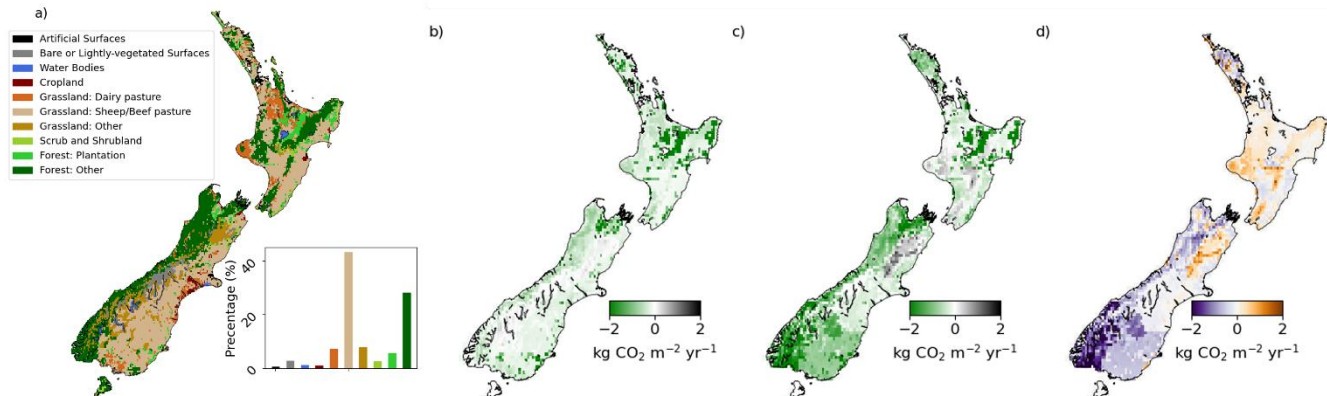

**Figure 8. Land type classifications used to map the Biome-BGCMuSo and CenW biomes across New Zealand, with the total land surface area contribution of each biome (a), 2011-2020 average prior (b), posterior (c) fluxes and their difference (d, posterior-prior). The spatial distribution of the posterior fluxes was constructed based on the prior flux maps.**

The north-western and central-western regions in the South Island (regions 9 and 11) are characterised by a net sink for all years, impacted by the forest activity in this area (mixture of mainly indigenous and some exotic forests). We observe a strong sink signal in New Zealand's southernmost regions as well (region 13). A large part of this region is covered by mature, indigenous forests (Fig. 8a). Region 13 is the largest region in our model (Table S3), leading to the strongest region-based sink signal. However, the area-based estimates are within uncertainties in comparison with region 9 (Table S4-S7). Hereinafter, as in Steinkamp et al. (2017) we will also refer to region 13 as Fiordland, although it includes regions that are not part of the Fiordland National Park (i.e., western Southland, Stewart Island and western Otago).

Overall, the location of the stronger posterior $CO_2$ sink follows a similar pattern as found in Steinkamp et al. (2017). A large portion of the sink in the posterior estimates are in the South Island. However, where Steinkamp et al. (2017) found a strong sink localised in the southwest of the South Island (region 13), we find a much more spatially distributed sink. Although the sink is largest in region 13, other regions in the southern half of the South Island also show larger carbon uptake than the prior (regions 9, 11, 15). Averaged over the 2011-2020 period, the posterior fluxes suggest a total uptake of -111 ± 26 Tg $CO_2$ yr$^{-1}$ in the South Island (69 Tg $CO_2$ yr$^{-1}$ stronger uptake from the prior) and -60 ± 16 Tg $CO_2$ yr$^{-1}$ in the North Island (17 Tg $CO_2$ yr$^{-1}$ weaker uptake from the prior). The area-based posterior flux estimates suggest a -0.64 ± 0.15 kg $CO_2$ m$^{-2}$ yr$^{-1}$ sink activity for the South Island and -0.42 ± 0.11 kg $CO_2$ m$^{-2}$ yr$^{-1}$ for the North Island, a stronger sink activity from the prior South Island estimates (-0.24 ± 0.09 kg $CO_2$ m$^{-2}$ yr$^{-1}$) and weaker for the North Island (-0.54 ± 0.08 kg $CO_2$ m$^{-2}$ yr$^{-1}$). However, we note that our measurement network has lower sensitivity to the northern part of the North Island (Steinkamp et al., 2017), hence additional and stronger sink or source regions can appear when adding measurements from additional sites into our inversion system (in case of a biased prior flux assumption).



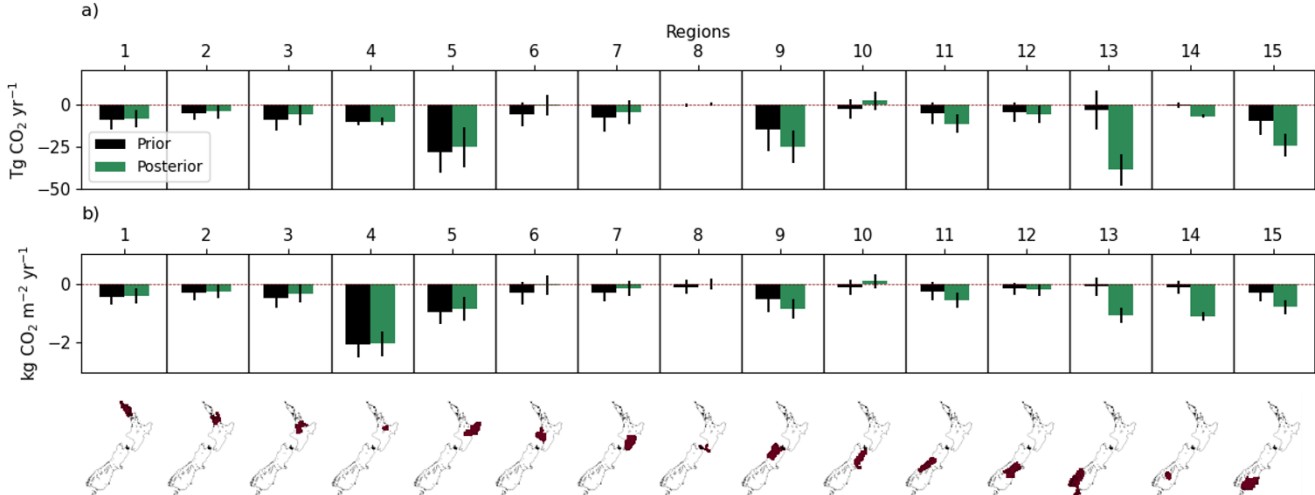


**Figure 9. 2011-2020 average CO₂ prior (black) and posterior (green) net air-land flux estimates for the 15 inversion land regions in units of Tg CO₂ yr⁻¹ (a) and area-based flux estimates in kg CO₂ m⁻² yr⁻¹ (b). The error bars represent the standard deviation. The inversion land regions are designed to also include small local regions around the measurement sites (i.e., Baring Head: region 8; Lauder: region 14). The purpose of these regions is to capture the local CO₂ exchange signal (Steinkamp et al., 2017).**

**3.1 Seasonal variability**

The posterior fluxes are characterised by stronger springtime uptake and weaker winter emissions compared to the prior (Fig 10, Fig. 11 and Fig. 12). The stronger posterior uptake in spring suggest that the peak of the growing season is potentially occurring earlier relative to prior estimates. However, in Sect. 4.3 we discuss a potential impact of the $CO_2$ diurnal cycle bias on the spring/summer estimates. Conversely, the posterior fluxes in winter and autumn show robust

discrepancies relative to the prior estimates. On average winter periods in the posterior fluxes suggest a weaker net source, while during autumn we find a neutral $CO_2$ exchange, suggesting suppressed or offset respiration by additional $CO_2$ uptake due to plant activity during these periods.

The regions in the South Island that showed larger $CO_2$ uptake than the prior estimates (regions 9 and 13, Fig. 11 and S12), all show reduced net $CO_2$ flux into the atmosphere (e.g., suppressed respiration) during autumn/winter, suggesting

$CO_2$ uptake throughout the year. The weak autumn/winter $CO_2$ net source is most pronounced in region 9, which is dominated by indigenous forests. Similar to region 9, the Fiordland area (region 13) also shows only weak source activity during autumn/winter periods. Although traditionally it has been assumed that mature forests are almost carbon neutral (Kira & Shidei, 1967; Odum, 1969; Luyssaert et al., 2008), our results suggest that these environments can potentially have significant carbon uptake. The low autumn/winter $CO_2$ release was not evident in Steinkamp et al. (2017), possibly due to

the shorter inversion time period (2011-2013), measurement gaps, and strong drought conditions during 2012 and 2013 that released larger amounts of $CO_2$ to the atmosphere.





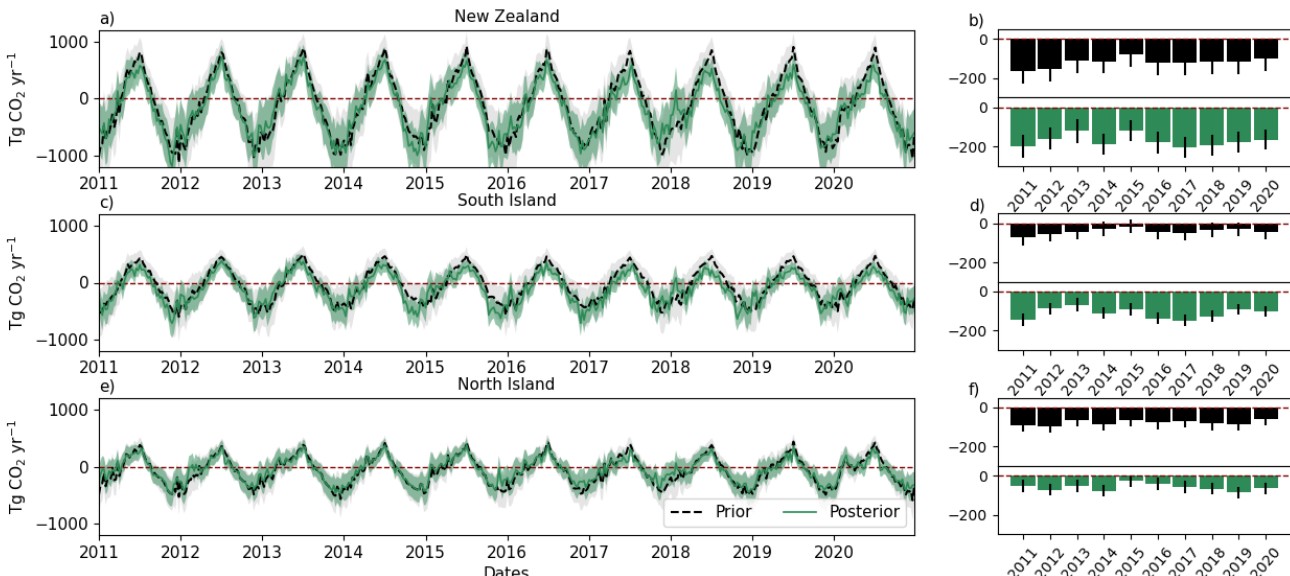

**Figure 10. Weekly time series (a, c, e) and annual (b, d, f) CO₂ prior (black) and posterior (green) net air-land flux estimates for New Zealand (a,b), the South Island (c,d) and North Island (e,f).**

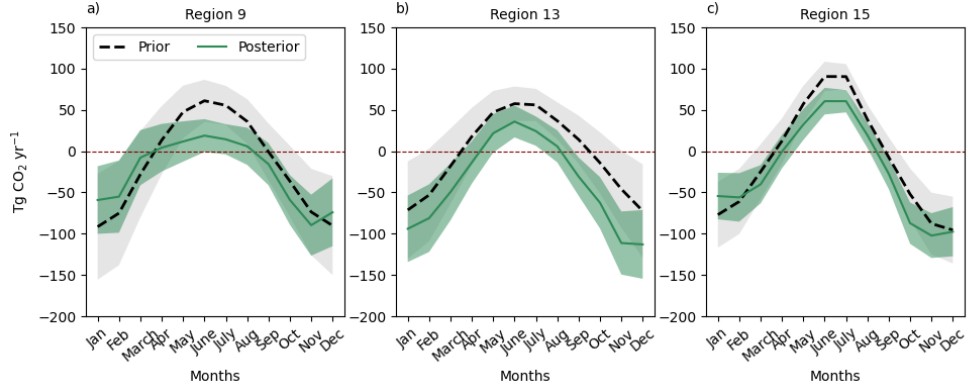


**Figure 11. 2011-2020 average monthly CO₂ prior (black) and posterior (green) net air-land flux estimates for selected regions.**



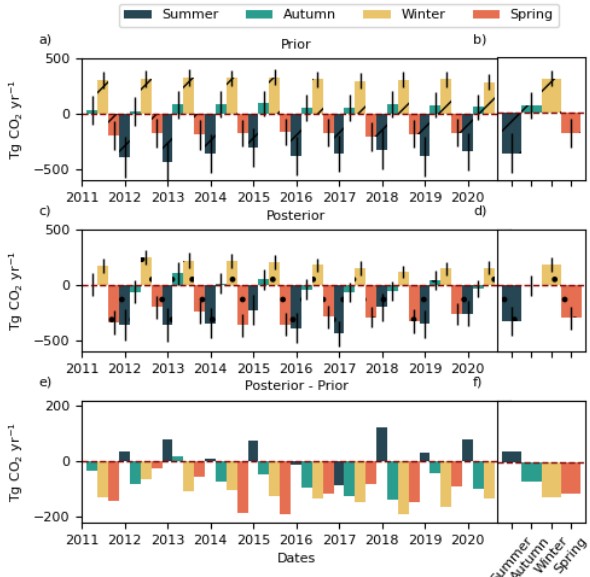

**Figure 12. Mean South Island annual seasonal (summer – December to February, autumn – March to May, winter – June to August, spring – September to November) CO₂ prior (a), posterior (c) net air-land flux estimates and their difference (e). Subplots b), d) and f) show the 2011-2020 average values for each season. The first and last season (summer) is removed from the plot and calculation due to insufficient number of months to calculate the seasonal average. All the regions can be found in Figs. S13-S14.**

## 3.2 Interannual variability

Steinkamp et al. (2017) found stronger $CO_2$ uptake during 2011-2013 than that report in the Inventory and the prior bottom-up model estimates (Biome-BGC). However, the posterior $CO_2$ fluxes over the three years showed a decreasing trend, potentially pointing to a transient sink. The decreasing trend was heavily influenced by measurement gaps and drought conditions, and a longer inversion period was needed to ascertain whether the inferred stronger posterior $CO_2$ sink was real and sustained over a longer observation period.

Based on our decade-long inversion, we found that the sink observed between 2011-2013 did not diminish in subsequent years, thus confirming that the atmospheric $CO_2$ signal supports the existence of an overall stronger national scale $CO_2$ uptake than reported in the 2024 Inventory and the prior fluxes. We find differences in the year-to-year changes of the prior and posterior fluxes in both the South and North Island (Fig. 10b,d,f). The most pronounced interannual variability in the posterior fluxes are in regions 13 and 15 (Fig. S12); however, we are cautious in interpreting the interannual variability due to additional uncertainties that impact the flux estimates. These include the impact of measurement data gaps (Fig. 2c,d), that will tend to keep the posterior fluxes closer to the prior flux estimates, as well as the impact of changes in the transport model that will be discussed in Sect. 4.



## 4 Sensitivity and validation

Uncertainties in top-down flux estimates on a regional to national scale are caused by a number of factors including 1) data availability in the country-wide measurement network (Berchet et al., 2013; Kountouris et al., 2018); 2) challenges in defining an appropriate background (Göckede et al., 2010); 3) biases in the prior fluxes (Peylin et al., 2011; Saeki & Patra, 420 2017; Sajeev et al., 2019); 4) atmospheric transport model and mixing errors (Baker et al., 2006; Prather et al., 2008); 5) aggregation errors (Kaminski et al., 2001; Turner & Jacob, 2015); and 6) the specific technical setup and assumptions in the inversion such as Gaussian assumptions (Miller et al., 2014) or representation of the $CO_2$ diurnal cycle (Gerbig et al., 2003; White et al., 2019). We conducted a suite of tests to assess the sensitivity of our results to specific setups and assumptions in our inversion system and used different diagnostics (i.e., residuals, Degrees of Freedom, Averaging) to quantify and better 425 understand the uncertainty and possible biases in the performance of the inversion.

### 4.1 Transport model

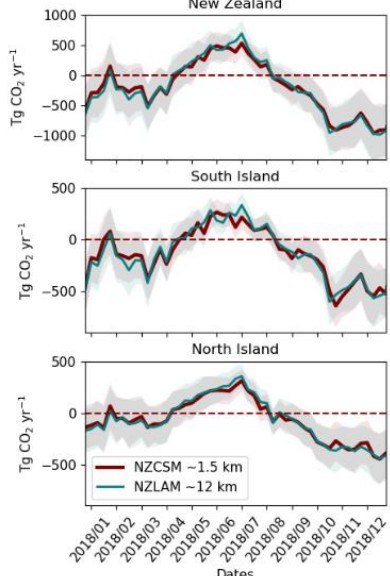

**Figure 13. Weekly posterior flux estimates using NZLAM (≈12 km) and NZCSM (≈1.5 km) meteorological input for year 2018.**

We compared our inversion results using NZCSM (≈1.5 km) and NZLAM (≈12 km) inputs for the NAME III model to 430 identify the impact of the resolution of the transport model on the posterior $CO_2$ fluxes. We ran the inversion for 2018 when both model outputs were available.

The differences in the posterior flux estimates were more pronounced for the individual inversion regions (Fig S15), but these differences averaged out when aggregated to larger regions (Fig. 13). On average, using the updated ≈1.5 km



NZCSM input led to a slight additional 2.3 Tg $CO_2$ yr$^{-1}$ increase of the sink for New Zealand as a whole (South Island: 0.5
435   Tg $CO_2$ yr$^{-1}$, North Island: 1.8 Tg $CO_2$ yr$^{-1}$). The Fiordland region showed a 10 Tg $CO_2$ yr$^{-1}$ difference in the posterior
fluxes when using different meteorological input, a reduced sink when using NZCSM (Fig S14). In Sect. 4.3 we identify
additional changes in the posterior fluxes, driven by the changeover between transport models that was not captured in the
2018 data.

## 4.2 Prior and background estimates

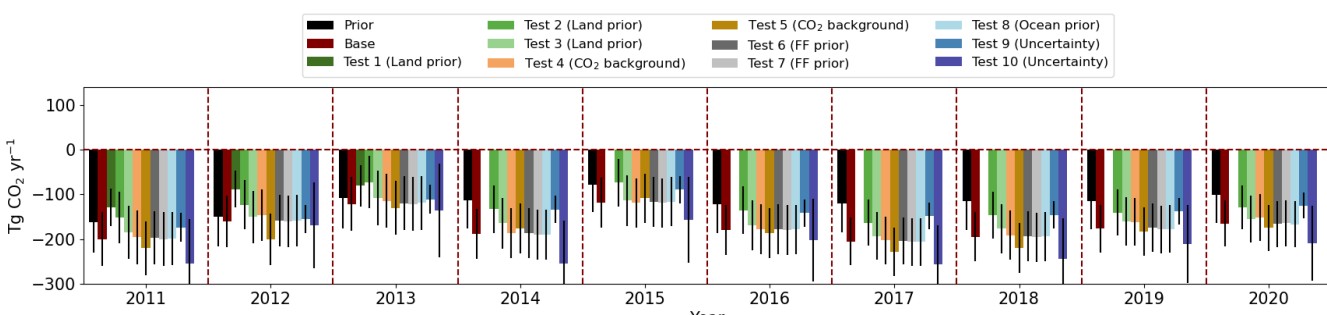


**Figure 14. Annual posterior flux estimates from the sensitivity tests, as well as the prior (black) and posterior (red) estimates from our base inversion. A detailed description of the setup for each test can be found in Table 2.**

The estimated posterior fluxes are strongly dependent on the choice of the prior land fluxes as well as their estimated
uncertainties. We compared our base inversion (using combined Biome-BGCMuSo and CenW fluxes) with the Biome-BGC
fluxes and land cover-map from Steinkamp et al. (2017) (Test 1, Fig. 14, Fig. S16). Updating the prior land fluxes and land
cover-map resulted in a 61 ± 14 Tg $CO_2$ yr$^{-1}$ increase in the national $CO_2$ sink, representing a 73% change in the posterior
fluxes. These results are additionally impacted by the weaker measurement constraint in the North Island (i.e., the posterior
estimates will be strongly impacted by the prior). Next, we tested an inversion setup where we retained the Biome-
BGCMuSo evergreen needleleaf forest biome category for the plantation forest category instead of the CenW pine fluxes
(Test 2). Using the Biome-BGCMuSo evergreen needleleaf forest fluxes instead of CenW pine fluxes led to a 44 ± 6 Tg $CO_2$
yr$^{-1}$ sink reduction in the posterior fluxes. As shown in Fig. 7, this is expected due to the all-year-round $CO_2$ uptake in the
CenW pine fluxes, which is not present in the Biome-BGCMuSo evergreen needleleaf forest fluxes. We also tested the land
cover-map introduced in Steinkamp et al. (2017) with the updated prior fluxes in our base inversion (Test 3). Using the older
land cover-map described in Steinkamp et al. (2017) led to a 13 ± 5 Tg $CO_2$ yr$^{-1}$ sink reduction in the posterior fluxes.
Reducing the prior land uncertainties in our base inversion by a half (Test 9) reduced the posterior $CO_2$ sink due to a tighter
constraint on the prior fluxes, while doubling the uncertainties (Test 10) further increased the $CO_2$ sink in regions that were
well observed by the measurement network.





**Table 2. Inversion sensitivity tests\*. Test 9 and 10 (not shown in the table) are the same as the base setup but with half and double the prior land uncertainty. All simulations used NZLAM input for 2011-2013 and NZCSM for 2014-2020.**

|  | Land cover | Land prior | Ocean prior | Fossil fuel prior | Background $CO_2$ |
|---|---|---|---|---|---|
| Base | LCDB v5.0 | Biome-BGCMuSo + CenW | Landschützer | EDGAR v7, scaled** | BHD + TF5*** |
| Test 1 | LCDB v5.0 | Biome-BGC | Landschützer | EDGAR v7, scaled** | BHD + TF5*** |
| Test 2 | LCDB v5.0 | Biome-BGCMuSo | Landschützer | EDGAR v7, scaled** | BHD + TF5*** |
| Test 3 | LCDB v3.0 | Biome-BGCMuSo + CenW | Landschützer | EDGAR v7, scaled** | BHD + TF5*** |
| Test 4 | LCDB v5.0 | Biome-BGCMuSo + CenW | Landschützer | EDGAR v7, scaled** | BHD*** |
| Test 5 | LCDB v5.0 | Biome-BGCMuSo + CenW | Landschützer | EDGAR v7, scaled** | CarbonTracker |
| Test 6 | LCDB v5.0 | Biome-BGCMuSo + CenW | Landschützer | EDGAR v4.2 | BHD + TF5*** |
| Test 7 | LCDB v5.0 | Biome-BGCMuSo + CenW | Landschützer | EDGAR v7 | BHD + TF5*** |
| Test 8 | LCDB v5.0 | Biome-BGCMuSo + CenW | Takahashi | EDGAR v7, scaled** | BHD + TF5*** |

*NZLAM for years 2011-2013 and NZCSM for years 2014-2020. Not all tests were performed for the whole inversion period (2011-2020) due to the
lack of year specific data for certain sensitivity tests.

** scaled refers to scaling the values to the 2023 Inventory estimates

***BHD: Baring Head, TF5: Trans Future 5

Our sensitivity tests suggest the choice of background has little impact on the inverse estimates for New Zealand, which is likely due to its location far removed from most terrestrial sources and sinks. We trialled three different choices of background: our base inversion based on steady interval data at Baring Head combined with ship-based observations, $CO_2$ from Baring Head only (Test 4), and CarbonTracker $CO_2$ estimates (version CT2022, Jacobson et al. (2023)) (Test 5). Using Baring Head only decreased the posterior sink ($6 \pm 5$ Tg $CO_2$ yr$^{-1}$), while using CarbonTracker increased the sink ($12 \pm 15$ Tg $CO_2$ yr$^{-1}$). The differences in the posterior fluxes from all other sensitivity tests were also small. We found low sensitivity to the choice of the anthropogenic prior flux fields (Tests 6 and 7). We have tested different ocean prior fluxes (Test 8) to highlight potential $CO_2$ transfer from the land to the coastal regions and ocean but found low sensitivity to the choice of ocean priors as well. In summary, using updated priors resulted in an increase of the posterior sink, and increasing the prior uncertainties further increased the posterior sink.

## 4.3 Diurnal variability

We performed an additional sensitivity test to assess the implications of the lack of the $CO_2$ diurnal cycle in our prior land fluxes. Due to the use of weekly prior land fluxes, time integrated footprints and afternoon $CO_2$ measurements, the modelled terrestrial diurnal cycle could not be fully resolved. Omitting night-time measurements and hourly prior land $CO_2$ fluxes could potentially result in overestimated negative posterior fluxes (i.e., sink) due to the exclusion of the $CO_2$ diurnal cycle.





We excluded night-time measurements due to conditions such as lower wind speeds, weak vertical mixing and a shallow PBL. Measurements collected during these conditions primarily reflect local $CO_2$ exchange processes and are not representative of process at larger regional scales. Further, both Biome-BGCMuSo and CenW simulate biospheric fluxes at a

daily native temporal resolution.

We performed an Observing System Simulation Experiment (OSSE) to investigate the impact of using afternoon measurements with weekly prior land prior fluxes in our base inversion system. Our experiment used a synthetic $CO_2$ dataset from Baring Head and Lauder at 13:00-14:00 and 15:00-16:00 local time that had been created with dis-aggregated hourly footprints and hourly prior land fluxes. Since the prior models do not simulate the $CO_2$ diurnal cycle, we used highly

idealised hourly fluxes as described in Steinkamp et al. (2017). We assigned the same prior flux and measurement uncertainties to the synthetic dataset as in our base inversion. The experiment used the same land prior fluxes as the ones used for the synthetic data creation but was averaged to weekly values (i.e., excluding the diurnal cycle). The difference between the prior (i.e., true) and posterior estimates (and the ability of the posterior to recover the "true" diurnal cycle) was expected to highlight the impact of the temporal resolution in the land prior (i.e., lack of a diurnal cycle) on our inversion

results. However, this test does not reflect the impact of the inclusion of night-time measurements in the inversion system (Sect. 7).

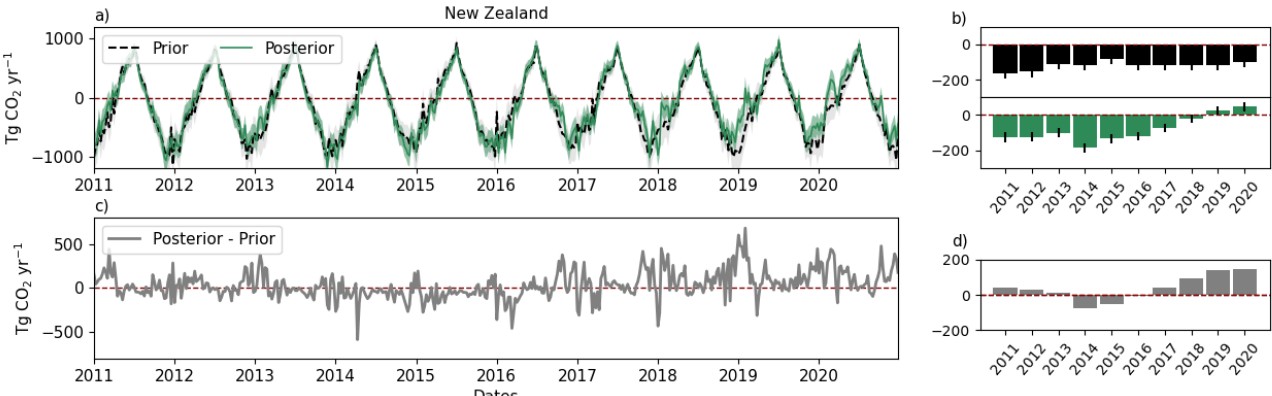

**Figure 15. Weekly (a) and annual (b) $CO_2$ prior (black) and posterior (green) net air-land flux estimates for New Zealand from the diurnal cycle test. The difference between the posterior and prior estimates is shown on c) and d) Tg $CO_2$ yr$^{-1}$.**

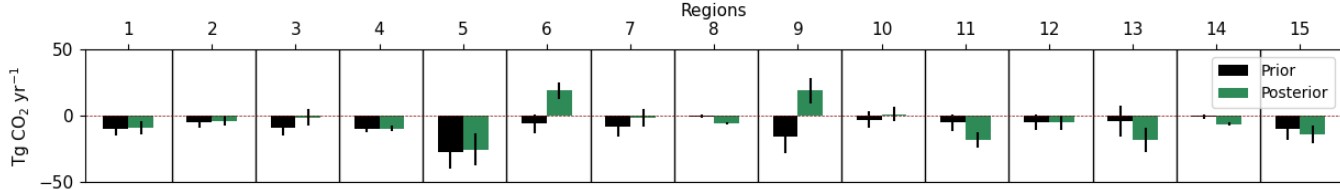


**Figure 16. 2011-2020 average $CO_2$ prior (black) and posterior (green) net air-land flux estimates for the 15 inversion land regions from the diurnal cycle test.**



Identical prior and posterior fluxes would suggest that there is no bias in our inversion system, while any difference between the two values points to a potentially over- or underestimated $CO_2$ sink or source. Averaged over all of New Zealand, we did not see a consistent offset between the prior and posterior fluxes (Fig. 15); instead, the results suggest both over- and under-estimated $CO_2$ fluxes during certain time periods. The summer/spring periods point to a potential diurnal cycle bias; however, this bias is not present during autumn/winter periods when our results suggest suppressed respiration (Fig. 15 and S19).

We find a strong interannual variability of the diurnal cycle bias which is impacted by the transition between different meteorology input fields in our transport model, described in Sect. 2.2. The impact of the transport model on the posterior fluxes is highlighted in the regional plots shown in Fig. S17. The diurnal cycle test suggests that, on average, the strong sink observed in the Fiordland (region 13) and Southland region (region 15) is overestimated (Fig. 16) and underestimated along the West Coast (region 9). However, the results for Fiordland and Southland suggest a mixture of both over- and under-estimated $CO_2$ fluxes, with an overestimated sink during 2011-2013, when NZLAM was used, while for later years the overestimated sink is less pronounced (Fig. S17). We note that these results are based on highly idealized hourly prior fluxes that can introduce additional uncertainties in the estimated bias. In the North Island, our inversion system tends to underestimate the sink and points to reduced uptake during spring/summer periods (Fig. S18). A $37 \pm 26$ Tg $CO_2$ $yr^{-1}$ difference exists between the prior and posterior fluxes. Biases in the South Island, on average suggest on underestimated sink of $1 \pm 48$ Tg $CO_2$ $yr^{-1}$. Similar to the South Island regions the bias towards an overestimated sink is more pronounced for the earlier record of the inversion and this bias was mitigated after transport model improvements were made. Overall, in the context of an overestimated sink, the sensitivity to the diurnal cycle were less pronounced for later years as the NZCSM model improved (Sect. 2.2).

## 4.4 Inversion performance

Analysis of the residuals (model - observation) (Fig. 17) contains information about potential biases impacting the posterior fluxes. Residuals represent the differences between the modelled and measured $CO_2$ mole fractions, with the modelled values being the optimized $CO_2$ mole fractions by propagating the posterior flux estimates through the inversion. A positive mean bias would suggest that the modelled $CO_2$ values are higher relative to the measurements and that the inversion struggles to reproduce the low $CO_2$ observations, while a negative bias would suggest the opposite, i.e., lower modelled $CO_2$ values relative to measurements.

We find a positive bias in our base inversion for both sites and measurement times. Baring Head shows similar residual values between the 13:00-14:00 and 15:00-16:00 local time with a bias of 0.19 ppm. The residuals at Lauder show a higher mean bias, 0.59 ppm at 13:00-14:00 and 1.39 ppm at 15:00-16:00. We also find a small temporal/seasonal pattern of the residuals, with higher values during winter and lower values during summer periods. As discussed in Steinkamp et al.



(2017) there is a strong indication that the biases in the residuals are a product of biases between measured and modelled

PBL depth or biases from excluding the diurnal variability of $CO_2$ in the data (by using afternoon data only) and prior fluxes.

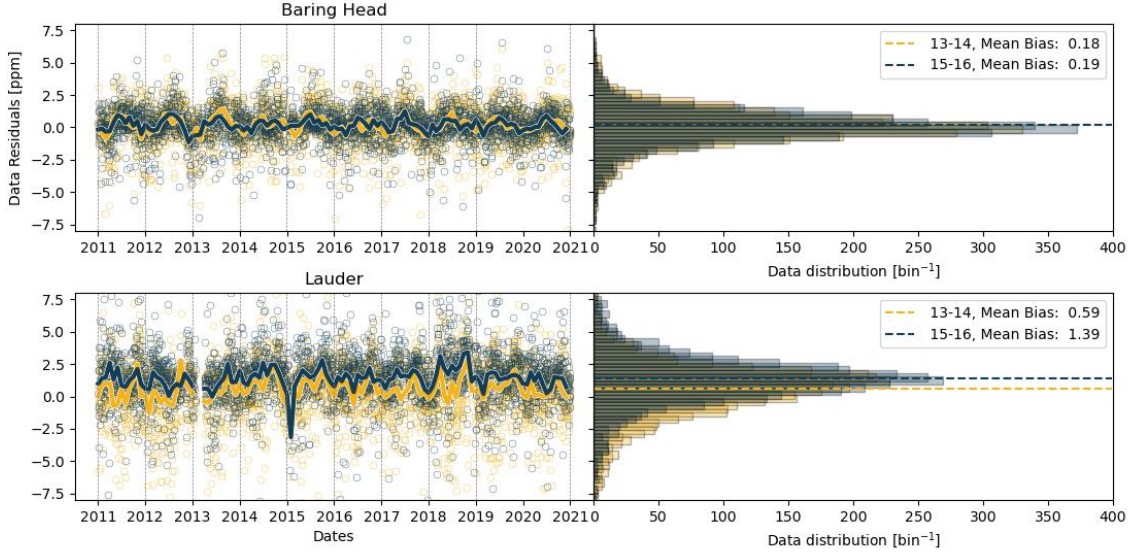

**Figure 17. Modelled and measured CO₂ differences (residuals) at 13:00-14:00 and 15:00-16:00 local time at Baring Head (top) and Lauder (bottom) for 2011-2020. The left plots show the residuals for each day (circles) and seasonal cycle of the residuals (solid lines). The right plots show the residual distribution with the mean bias (dashed line).**

Regional Degrees of Freedom (DOFs, Fig. 18a,b) and Averaging Kernel (AK, Fig. 18c and S19) (Rodgers, 2000) provide information about the ability of our measurement system to constrain the fluxes. The DOFs are the trace of AK and they provide a quantitative measure of the number of independent pieces of information in the constrained fluxes, provided by the measurements. Based on the monthly covariance matrices, each region can have a maximum of 120 DOFs for the whole inversion period (i.e., 12 per year, 10 inversion years). Regions with higher DOFs suggest stronger flux constraints.

Our results suggest that the South Island regions have greater measurement sensitivity than the North Island regions, and are better constrained by the current observational network, as indicated by higher DOF values, which signify stronger observational influence on the inversion results. The DOFs exhibit seasonal variation, with certain regions, particularly in the South Island, generally owing higher DOFs during austral summer periods. Winter periods and regions in the North Island, especially in central and northern areas, have lower DOFs, indicating a greater reliance on prior emissions estimates due to

weaker observational constraints. The mean DOF map further supports this, showing higher sensitivity in the South Island (Fig. 18c).





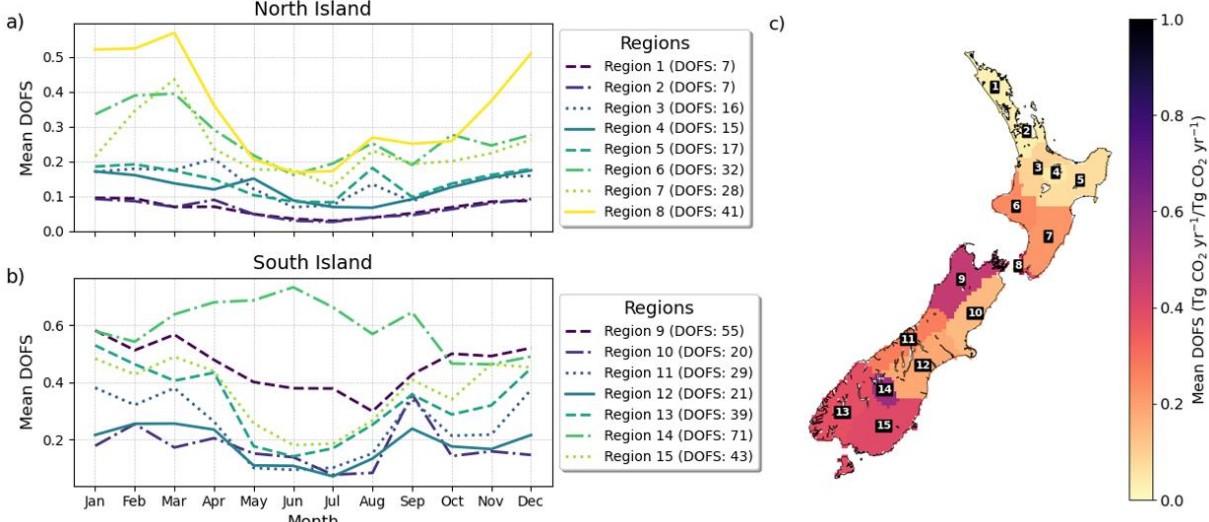

**Figure 18. Monthly mean Degrees of Freedom (DOFs) for each inversion region (a, b), with total DOFs for each region shown in the legend. The full timeseries for each region is shown in Fig. S20. Subplot c) shows the average 2011-2020 DOFs for each inversion region, with labelled region numbers.**


The low off-diagonal elements of the covariance matrix (Fig. S21) suggest that the different regional posterior fluxes can be individually constrained. The Fiordland inversion region on average (region 13) showed a stronger correlation with region 15, suggesting that the larger posterior sink in region 15 is potentially influenced by region 13.

# 5 CO₂ exchange processes

## 5.1 Understanding differences between the prior and posterior

The main difference between our posterior and prior estimates is the suppressed autumn/winter respiration in the posterior (Fig. 10). Both features are strongest in regions that are dominated by forests and well constrained by our observational network (i.e., South Island). The strongest sink occurs in regions dominated by mature indigenous forests, which suggests that these forests may be a more efficient carbon sink than suggested by the terrestrial biosphere model. Common
assumptions in bottom-up terrestrial biosphere models may play a role in the resulting differences between the posterior and prior estimates. These include the response of the model to erosion and landslide disturbance, impact of drought and freezing, modelling of animal respiration, biases in forest model parameters, representation of harvest and replanting cycle.

Most models, such as Biome-BGCMuSo, do not accurately represent rapid accumulation of organic matter in vegetation and soils nor burial and preservation following erosion and/or landslide disturbance (Stallard, 1998; Dymond,
2010; Berhe et al., 2018). Further, seasonal patterns of respiration could be overemphasised in Biome-BGCMuSo





simulations because the model parameters have been developed for highly seasonal temperate environments with regular freezing and/or drought. This could result in overestimates of respiration in the physiologic responses of decomposer organisms to drought or freezing (Schnecker et al., 2023); however, these are expected to be infrequent events in the New Zealand regions where significant sinks relative to the priors have been found.

Further, a possible explanation for the discrepancy between the prior and posterior values in grassland regions is animal respiration (i.e., dairy cows respire about 50 % of the carbon they eat, Sect. 5.2.2), which is not included in the prior estimates. This would impact the main agricultural regions, leading to a reduced net sink or a small source in the prior. We observe this correction of the prior in major agricultural/animal regions in the North Island (region 3, 6, 7), as well as region 10 in the South Island. Regions 12 and 15 are also strong agricultural regions; however, we do not observe this correction,

presumably due to biases in other carbon exchange processes.

In the Biome-BGCMuSo model, the evergreen needleleaf and broadleaf forest categories (planted and indigenous forests) are not optimised for New Zealand. The model uses the default parameters that were tuned to Northern Hemisphere forests (White et al., 2000; Pietsch et al., 2005; Hidy et al., 2022). The CenW model provides pine flux estimates (representative of plantation forests) that were optimised for New Zealand conditions (Kirschbaum & Watt, 2011). Figure 19

shows the average 2011-2020 NEE, GPP and ER seasonal cycle, as well the mean 2011-2020 contribution from the evergreen broadleaf forest, evergreen needleleaf forest and CenW pine categories, mapped based on the landcover map shown in Fig. 8a.

There are notable differences between the exotic forest flux estimates in the two models. The CenW pine fluxes show year-round net uptake while the Biome-BGCMuSo evergreen needleleaf forest fluxes suggest a seasonal cycle with

uptake during summer periods and respiration during winter (Fig. 19a). The Biome-BGCMuSo evergreen needleleaf forest category shows a similar annual flux magnitude between GPP and ER (Fig. 19e) but with a stronger production, resulting in an overall net -12 Tg $CO_2$ yr$^{-1}$ sink (Fig. 19d). Note, however, that the evergreen needleleaf forest fluxes modelled with Biome-BGCMuSo do not include the effects of the harvest and replanting cycle typical of managed plantation forests (due to computational limitations the harvest module in Biome-BGCMuSo was not activated). Plantation forests tend to have greater

GPP relative to ER, as harvested material is generally transported elsewhere before it starts decomposing and releasing $CO_2$, and young replanted forests grow and accumulate carbon rapidly. A significant portion of New Zealand's raw wood products are exported (12.1 ± 3.7 Tg $CO_2$ yr$^{-1}$; Villalobos et al. (2023)), so if the harvested carbon is eventually released to the atmosphere, it would not be detected in measurements over New Zealand. The Biome-BGCMuSo evergreen needleleaf forest fluxes are more representative of mature, unmanaged pine forest. The CenW model suggests larger differences

between GPP and ER, with stronger GPP, resulting in an annual net sink of -61 Tg $CO_2$ yr$^{-1}$ (2011-2020 average, Fig. 19d) for the pine forested areas, and year-round net uptake. Winter temperatures in New Zealand are generally mild, hence allowing year-round plant activity. Moreover, other studies have also shown that winter conditions in New Zealand can lead to year-round productivity under other vegetation covers (Campbell et al., 2014), and the high productivity of pine forests in New Zealand is consistent with extensive forest-inventory data in the country (Kirschbaum & Watt, 2011).





The Biome-BGCMuSo evergreen broadleaf forest category shows an NEE seasonal cycle that is similar to the evergreen needleleaf forest category, but with a weaker amplitude (Fig. 19a). Averaged for 2011-2020, it results in a -26 Tg $CO_2$ yr$^{-1}$ net uptake (Fig. 19d). Note, this uptake is not directly comparable to the evergreen needleleaf forest fluxes since native forests cover a larger area across New Zealand (Fig. 8). Mature indigenous forests (i.e., pre-1990 natural forests) cover 29% of New Zealand, while planted exotic forests cover about 8% of the country (MfE, 2024). The area-based

estimates of each forest flux suggest a net uptake of -0.9 kg $CO_2$ m$^{-2}$ yr$^{-1}$ for evergreen needleleaf forests, -0.3 kg $CO_2$ m$^{-2}$ yr$^{-1}$ for evergreen broadleaf forests and a significantly stronger uptake of -4 kg $CO_2$ m$^{-2}$ yr$^{-1}$ from the CenW pine fluxes (averaged for 2011-2020). Assuming that mature, indigenous forests can also take up $CO_2$ all year round, the large posterior sink in the South Island can be linked to high year-round productivity of these forests.

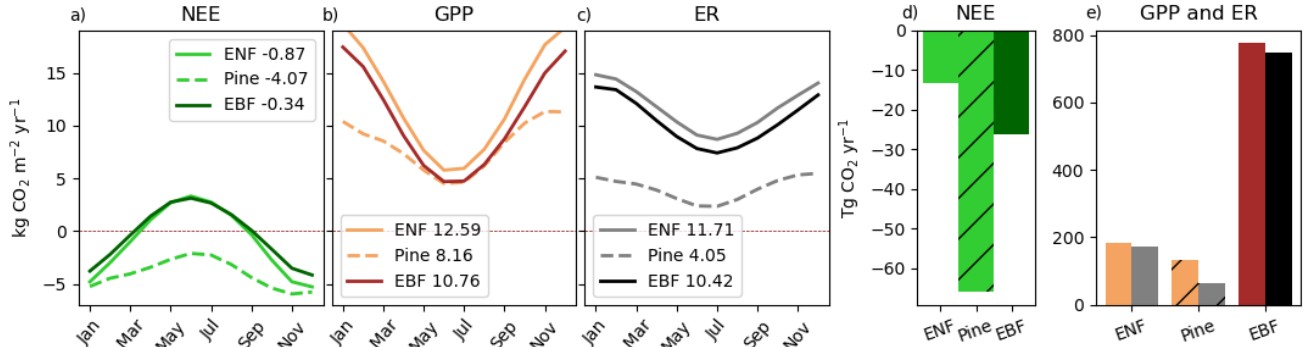

**Figure 19. Net ecosystem exchange (NEE, a), gross primary production (GPP, b) and ecosystem respiration (ER, c) seasonal cycle (2011- 2020 average, annual net estimates are shown in the legend), as well mean 2011-2020 contribution (d and e) from the Biome-BGCMuSo evergreen broadleaf forest (EBF) and evergreen needleleaf forest (ENF) categories and CenW pine category. All values are based on the contributions of each forest type for the forested regions shown in Fig. 8a.**

## 5.2 Bottom-up $CO_2$ flux estimates

We compare our $CO_2$ inversion posterior estimates with the current understanding derived from bottom-up estimates. This comparison is crucial as we recognize that there are significant discrepancies between the two approaches that require careful resolution.  To address these differences, we compare our inversion estimates with published data on forests, grasslands, along with $CO_2$ lateral transport, with particular attention to the dynamics of fjord environments. By doing so, we aim to better reconcile the two methodologies and enhance the accuracy of both top-down and bottom-up $CO_2$ flux assessments.

Reconciling atmospheric observations with bottom-up flux estimates can be divided into two broad categories.  The first is an imbalance of photosynthesis and respiration driving a net uptake into the land, in either forests or grasslands.  The second is lateral transport of carbon through rivers, soil erosion and landslides. While the carbon export itself cannot be observed by the atmospheric inversion system, such carbon export can allow additional $CO_2$ uptake into soils, "renewing"





the carbon pool within the soils. In the case of landslides, exposure of new surfaces results in development of new soils that
can sequester carbon. Thus, carbon export should result in apparent net uptake or sink in the regional atmospheric inversion.

### 5.2.1 Forests

Forests can store and sequester large amounts of carbon; hence their management can play a crucial role in climate change mitigation (United Nations, 1998; Griscom et al., 2017; Kirschbaum et al., 2024). In contrast to younger forests, mature forests are generally considered to be carbon neutral and stop accumulating carbon once the trees reach a certain
(species-dependent) age (Kira & Shidei, 1967; Odum, 1969; Sousa, 1984; Binkley et al., 2002; Holdaway et al., 2017). If mature forests are perturbed, however, tree growth can be stimulated again and lead to renewed carbon accumulation (Van Tuyl et al., 2005; Luyssaert et al., 2008; Stephenson et al., 2014; Brienen et al., 2015; Schimel et al., 2015; Holdaway et al., 2017). Pest control can further allow forest recovery and increased growth in native forest; however, the impact of these practices on the carbon balance are not well quantified. It was shown that the $CO_2$ fertilisation effect can lead to carbon
accumulation in young forests (Walker et al., 2019); however, its impact on mature forests is still being questioned (Jiang et al., 2020). Widely used models produce results which vary widely and depend on how models implement the relationship between photosynthesis and nitrogen limitation (Arora et al., 2020). Specific meteorological and growing conditions can further impact the $CO_2$ exchange from different ecosystems leading to changes in their photosynthetic or respiratory activity (Campbell et al., 2014; Duffy et al., 2021).

Current studies are still subject to uncertainties and debates with contradictory and complex conclusions in terms of the nature and magnitude of $CO_2$ exchange from different forest types (Carey et al., 2001; Pregitzer & Euskirchen, 2004; Bonan, 2008; Erb et al., 2013; Pugh et al., 2019). Carbon capture of newly-planted stands of native species suggest an upper limit of -0.95 to -1.5 kg $CO_2$ m$^{-2}$ yr$^{-1}$ (Kimberley et al., 2014; Holdaway et al., 2017) depending species of the trees (i.e., Red Beach, Puriri, Kauri, Totara) and as high as -3 kg $CO_2$ m$^{-2}$ yr$^{-1}$ in short-term for mixed shrubs. For secondary and mature
forests, Holdaway et al. (2017) estimated gains of less than -0.5 kg $CO_2$ m$^{-2}$ yr$^{-1}$ for the first 40 years and subsequently at steady state. Paul et al. (2021) found that New Zealand's natural forests are in balance with sequestration rate of 0.2 kg $CO_2$ m$^{-2}$ yr$^{-1}$ for regenerating forests and close to zero for tall forest. The carbon gain from production forests (e.g., pine) can vary across the country from about -1.1 kg $CO_2$ m$^{-2}$ yr$^{-1}$ at the South Island West Coast to -3.5 kg $CO_2$ m$^{-2}$ yr$^{-1}$ in the Taranaki region (nationally average number of -1.9 kg $CO_2$ m$^{-2}$ yr$^{-1}$) over about 30 years before felling. Our top-down estimates for
the Fiordland inversion region (region 13), with 42% (Table S3) of the region being predominantly covered by mature native forests suggest an average flux of -1 ± 0.3 kg $CO_2$ m$^{-2}$ yr$^{-1}$.





### 5.2.2 Grasslands

Grasslands contribute to carbon uptake through plant growth, which is mostly balanced by carbon loss through either plant decomposition or animal respiration. There is a slight surplus of carbon fixation to be balanced by methane flux and export of animal products as milk, meat or wool. There can be an additional gain or loss corresponding to changes in soil organic carbon (SOC). The carbon balance in forests is mainly driven by changes in woody biomass, for grasslands; however, the balance is primarily driven by changes in SOC or leaf biomass carbon (Kirschbaum et al., 2020). Previous studies have estimated dairy grasslands to be near carbon neutral, with net uptake during springtime, dependent on different management regimes (Kirschbaum et al., 2020; Wall et al., 2024). For a small set of sites, Schipper et al. (2014) found preliminary evidence for SOC gains in New Zealand pasture hill country of -0.26 kg $CO_2$ $m^{-2}$ $yr^{-1}$, while flat land and tussocks (Schipper et al., 2017) were near to carbon neutral; however, they noted that these estimates are highly uncertain due to poor data coverage. Pasture on drained peats acts as a source of 1.8 kg $CO_2$ $m^{-2}$ $yr^{-1}$ (mainly Waikato and Southland region, Campbell et al. (2021)) neither of which are evident in the inversion. Some land management practices also lead to losses of carbon, e.g., irrigation contributes to the source by 0.18 kg $CO_2$ $m^{-2}$ $yr^{-1}$ but only for the first ~10 years after irrigation commenced (Mudge et al., 2021). Further, dissolved organic carbon leaching in grasslands was estimated as a sink of -0.018 kg $CO_2$ $m^{-2}$ $yr^{-1}$ (Sparling et al., 2016); however, this leached carbon is likely partially mineralised to $CO_2$ during transport through vadose zone and in groundwater. It is assumed that 0.3 kg $CO_2$ $m^{-2}$ $yr^{-1}$ of the pasture milk solids in New Zealand is exported (estimated from milk production and carbon content of milk) while export of pasture meat and wool is estimated at 0.07 kg $CO_2$ $m^{-2}$ $yr^{-1}$. Furthermore, assuming that New Zealand has ~4.8 million cows (DairyNZ, 2022) and that 50% of the carbon intake is being respired as $CO_2$ by dairy cows, we estimate the $CO_2$ emission from animal respiration to be ~17 Tg $CO_2$ $yr^{-1}$ (~0.054 kg $CO_2$ $m^{-2}$ $yr^{-1}$), and an additional ~12 Tg $CO_2$ $yr^{-1}$ (~0.037 kg $CO_2$ $m^{-2}$ $yr^{-1}$) from sheep and beef cattle. Top-down sink estimates of regions with predominantly grazed grassland were as high as -1.3 kg $CO_2$ $m^{-2}$ $y^{-1}$ using Southland as an example (Table S7).

### 5.2.3 Lateral transport and fjords

Lateral transport, erosion and deposition of organic material can be very important in montane regions and other steeplands when accompanied by rapid re-establishment of productive vegetation on the disturbed landscape (Stallard, 1998; Berhe et al., 2018). While the carbon export itself cannot be observed by the atmospheric inversion system, such carbon export can allow additional $CO_2$ uptake, "renewing" the long-lived carbon pools in soils and wood. In the case of landslides, exposure of new surfaces results in development of new soils that can sequester carbon. Thus, carbon export should result in apparent net uptake or sink in the regional atmospheric inversion.

Globally, about 8 Pg $CO_2$ $yr^{-1}$ of carbon is exported through lateral transport, with about half of this being returned to the atmosphere from inland waters (Regnier et al., 2022; Tian et al., 2023). This return of carbon from inland waters will



be included in the atmospheric inversion as a source within the same regions as the export occurs, and therefore can be ignored in this reconciliation. The remaining roughly 4 Pg $CO_2$ yr$^{-1}$ is exported into coastal margins, as dissolved inorganic carbon (DIC, ~50%), dissolved organic carbon (DOC, ~30%) and particulate organic carbon (POC, ~ 20%). It is thought that about half of this carbon is sequestered into sediment over the long-term, with the remainder being returned to the atmosphere in the coastal margins or open ocean (Regnier et al., 2022; Tian et al., 2023); this return to the atmosphere will occur outside of the regional inversion footprint, and thus will appear as a sink and is not further considered here.

New Zealand's long coastline relative to land area, dramatic topography, high rainfall, tectonic activity and changing land uses all contribute to proportionately large sediment export, contributing about 1.7% of global sediment export (Hicks et al., 2011). Three studies have estimated New Zealand's carbon export at 15-19 Tg $CO_2$ yr$^{-1}$, each with different partitioning of the carbon between POC, DOC and DIC (Scott et al., 2006; Dymond, 2010; Villalobos et al., 2023), with more details in Sect. S6. The South Island West Coast and Fiordland contribute about 65% of this carbon export (Scott et al., 2006). These studies all rely on modelling and scaling up of a modest number of observations, and do not always consider all three carbon pools, indicating that more research is needed to better constrain the magnitude of carbon export from New Zealand.

We have identified a range of carbon exchange processes in mature and production forests, grasslands, fjord environments, as well as the role of lateral transport that could contribute to the differences. Lateral transport of carbon could explain about half of the observed atmospheric inversion sink. Forest and grassland carbon imbalances could plausibly explain the remainder of the difference between top-down and bottom-up, especially when forest disturbance and pest control are considered.

## 6. New Zealand's Greenhouse Gas Inventory

Estimates of greenhouse gas emissions and uptake from both the Inventory and top-down methods are subject to discrepancies and uncertainties (Baker et al., 2006; Göckede et al., 2010; Peylin et al., 2011; Saeki & Patra, 2017; Kountouris et al., 2018; Philip et al., 2019; Bastos et al., 2020). Closing the gap between inventory and top-down methods, and the need for the inclusion and improvement of top-down approaches for estimating carbon fluxes is recognised globally as a pivotal task for future development (IPCC, 2019).

In New Zealand's 2024 Inventory, compiled by the Ministry for the Environment (MfE), the net carbon uptake estimates from the land-use and forestry sector are based on land-use maps, providing activity data to which emission factors are applied to estimate carbon fluxes. Forest carbon fluxes are modelled, based on measurements from a representative network of forest plots that are scaled up to the national scale (MfE, 2024). National greenhouse gas inventories are focussed on anthropogenic change, rather than capturing all carbon fluxes as is the case for inverse modelling. However, in New





Zealand, all land and forests meet the IPCC definition of managed land (MfE, 2024), leading to an easier comparison between methods.

The 2024 Inventory estimated New Zealand's gross $CO_2$ emissions from all sectors excluding land use, land-use change and forestry for 2011-2020 to be 34 - 37 Tg $CO_2$ yr$^{-1}$ (MfE, 2024). The land use, land-use change and forestry (LULUCF, includes forest land, harvested wood products, cropland, grassland, wetlands, settlements and other land) sector was reported as an overall sink over this period, removing about 21 ± 14 - 29 ± 19 Tg $CO_2$ yr$^{-1}$ from the atmosphere. The largest fluxes were observed in forests (pre-1990 natural and planted forests, post-1989 natural and planted forests),

estimated to be a net sink of around 18 ± 12 Tg $CO_2$ yr$^{-1}$ in 2022, made up of 64 Tg $CO_2$ yr$^{-1}$ in removals offset by 46 Tg $CO_2$ yr$^{-1}$ in emissions (99% of which is from harvest), followed by a net gain in harvested wood products (of 7 Tg $CO_2$ yr$^{-1}$). Grasslands, croplands and other land areas were estimated as a net source adding around 5 Tg $CO_2$ yr$^{-1}$.

Our results have diverged further from the Inventory estimates than the earlier findings reported in Steinkamp et al. (2017). The inversion suggests a net -171 ± 29 Tg $CO_2$ yr$^{-1}$ uptake from New Zealand's terrestrial biosphere; however, these

results are not directly comparable with the Inventory (-24 Tg $CO_2$ yr$^{-1}$) due to differences between what the inventory reports and what the atmospheric measurements that underpin the inverse model detect. From the 46 Tg $CO_2$ yr$^{-1}$ forestry emissions, 30% is debris that decays away on site which is transferred to the deadwood pool while 70% exported off site and processed. Assuming that 70% of emissions occur outside of New Zealand, the net sink of forests would be estimated to be 55 Tg $CO_2$ yr$^{-1}$, instead of 18 Tg $CO_2$ yr$^{-1}$, which would partially close the gap. Further differences can be explained by the

regional variation in age class profiles that impact sequestration rates in production forestry, export of harvested wood from the site and the decay of harvested wood products, variance in the timing of the decay of harvest residues on site following tree harvest and natural mortality, agricultural exports, animal respiration, and assumptions that the above or below-ground grassland biomass is in steady state when in a grazing regime and that mineral soil carbon stocks are in steady state 20 years following a land use change as only the land use change impact on soil carbon is estimated in the inventory.

These differences may contribute further towards explaining the divergent results observed in this study. If a sink of this magnitude is occurring in mature natural forests, it should be reflected in the biomass and detected in the national forest plot monitoring programme, hence the results from the third tranche (Paul et al., 2021) of measurements in natural forests will provide a further useful step towards understanding where the observed sink is occurring.

**7 Conclusion**

Top-down regional and national scale estimates can provide crucial information to improve inventory and bottom-up methods and can help in identifying measurement limitations. We use inverse modelling to estimate New Zealand's carbon uptake and emissions using atmospheric measurements and model. This effort is part of the CarbonWatch-NZ research programme, which aims to develop a complete top-down picture of New Zealand's carbon balance using national inverse



modelling and targeted studies of New Zealand's forest, grassland and urban environments to support climate mitigation.
Our work here focuses on significant updates of a previously published atmospheric inverse modelling framework (Steinkamp et al., 2017) to constrain surface-atmosphere net $CO_2$ fluxes on a national scale.

Our decade-long (2011-2020) inverse modelling results point to a persistent national scale $CO_2$ net uptake across all New Zealand of -171 ± 29 Tg $CO_2$ $yr^{-1}$. Our estimates suggest a stronger national scale sink relative to both prior bottom-up and the independent Inventory estimates. We also find a stronger national scale uptake relative to Steinkamp et al. (2017),
primarily driven by an increased uptake in the prior flux estimates from regions covered by plantation forests in the North Island. However, we observe that our measurement network is less sensitive to the northern region of the North Island. As a result, incorporating measurements from additional sites into our inversion system could reveal stronger sink or source regions, particularly if there are biases in the prior flux assumptions.

We observe larger differences relative to prior bottom-up estimates in the South Island, with additional carbon
uptake in regions along the north-west and southern parts of the South Island, including the West Coast region, Fiordland and Southland (~53% of the national sink). Relative to Steinkamp et al. (2017) the sink in the South Island is more spread out between different regions. Southland is a grazed pasture region (~70% sheep and beef pasture) while large part of the West Coast and Fiordland regions is covered by mature indigenous forests (~50%), suggesting that these environments can potentially take up more carbon than thought before; however, the inversion estimates cannot be explained by current
understanding of these mature forests. We conducted sensitivity tests and inversion diagnostics to evaluate potential biases in our top-down estimates. The magnitude of the estimated $CO_2$ sink is moderately sensitive to modelling assumptions, specifically the choice of the prior terrestrial fluxes and modelling of the $CO_2$ diurnal cycle; however, across a wide range of sensitivity tests we still find a stronger sink relative to bottom-up estimates.

Our inversion system can estimate the net air–sea and air–land fluxes; however, it cannot identify the specific
processes driving the posterior fluxes. Relative to the prior bottom-up estimates, our results suggest that the stronger annual net $CO_2$ sink is impacted by suppressed autumn/winter respiration. The year-round $CO_2$ uptake by indigenous forests could explain a portion of the stronger sink in our posterior estimates. Further differences between the prior and posterior estimates can be driven by a combination of other carbon exchange processes such as: native forest regeneration after forest disturbance or pest control, changes in soil carbon and soil carbon recovery, lateral transport and carbon sequestration due to
erosion, deposition, burial due to strong tectonic activity and frequent landslides in some regions, marine productivity in the fjords and discharge of freshwater out of the fjords that export carbon from the Fiordland region to the open ocean and other.

While we have identified a number of $CO_2$ exchange processes that could be at play, we cannot explain the difference between top-down, bottom-up and inventory estimates. The best estimate of any individual process is not enough to explain the sink we see in our inversion; however, we acknowledge that uncertainties for some estimates can be large.
Region specific studies are required to identify if the observed $CO_2$ uptake is a permanent sink, as carbon transported laterally may be returned to the atmosphere, and additional work is needed to resolve the remaining differences between atmospheric measurements, terrestrial biosphere models, and the Inventory.





**Data availability**. The model code, input and output data has been made publicly available on Zenodo
(https://doi.org/10.5281/zenodo.14306816).

**Supplement**. The supplement related to this article is available online.

**Author contributions**. BB ran the inversion code, performed the analysis, and led the writing of the paper under the
supervision and guidance of SMF. The observational data set was prepared by GB, DS, RM and SN. The TF5 ship
measurements were provided by HM and SN. The prior Biome-BGCMuSo fluxes were provided by EDK, with support from
DH and ZB. CenW fluxes were provided by DLG, MUFK and LL. NAME III simulations were performed with guidance
from SM. WTB and JT contributed to the comparison of the inversion results to bottom-up estimates, while AB contributed
to the comparison with the Inventory. AG and DK provided input on the analysis of the inversion results. All authors
contributed to editing and revising the paper.

**Competing interests**. The authors declare that they have no conflict of interest.

**Acknowledgements**. The work presented here is part of the CarbonWatch-NZ project focusing on New Zealand's carbon
balance and carbon exchange processes. The authors are grateful to the CarbonWatch-NZ and NIWA Tropac and Lauder
research teams. We would like to acknowledge the funding schemes for supporting this research including MBIE and the
NIWA core funding through the Greenhouse Gases, Emissions and Carbon Cycle Science Programme. ZB and DH were
funded by the Hungary National Multidisciplinary Laboratory for Climate Change, RRF-2.3.1-21-2022-00014 project. We
acknowledge NIWA and Raghav Srinivasan for providing the VCSN data for the Biome-BGCMuSo and CenW model. The
author(s) wish to acknowledge the use of New Zealand eScience Infrastructure (NeSI) high performance computing
facilities, consulting support and/or training services as part of this research. New Zealand's national facilities are provided
by NeSI and funded jointly by NeSI's collaborator institutions and through the Ministry of Business, Innovation &
Employment's Research Infrastructure programme. URL https://www.nesi.org.nz. We deeply appreciate the generous
cooperation of Toyofuji Shipping Co., and Kagoshima Senpaku Co. with the NIES VOS program and would like to thank
the captain and crew of Trans Future 5. The atmospheric $CO_2$ data measured onboard Trans Future 5 are available on the
website https://soop.jp. We thank Scott Graham from Manaaki Whenua - Landcare Research for contributing with eddy
covariance data and John Hunt from Manaaki Whenua - Landcare Research for helpful discussions and contribution about
bottom-up $CO_2$ estimates and processes. We thank Peter Landschützer for consulting about the prior ocean fluxes.
CarbonTracker CT2022 results were provided by NOAA GML, Boulder, Colorado, USA from the website at
http://carbontracker.noaa.gov. EDGARv7.0 was obtained from https://edgar.jrc.ec.europa.eu/dataset_ghg70. The New



Zealand Material was also produced using Met Office Software. Lauder radiosonde data was provided by Ben Liley. We thank colleagues from New Zealand's Ministry for the Environment (MfE), for their input about the National Inventory.

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
