# Peer review of "Inverse modelling of New Zealand's carbon dioxide balance estimates a larger than expected carbon sink"

_EGUsphere, 2024_

## Author Response (AR1)

**Response to Reviewers**

**Reviewer 1**

We thank Reviewer #1 for their comments and suggestions on our manuscript "Inverse modelling of New Zealand's carbon dioxide balance estimates a larger than expected carbon sink". In the text below, we have included all the original reviewer comments and suggestions in black, followed by our response and relevant manuscript changes in red. Page and line numbers refer to marked-up manuscript version.

General comments:

This paper presents an update to a previous study (Steinkamp et al., 2017) that estimated New Zealand's $CO_2$ uptake from atmospheric $CO_2$ measurements. This new study provides a significant update, as it extends the original 3-year study to 10 years, it improves the resolution of the atmospheric transport by a factor of about 10 (which was important to test due to New Zealand's complex topography), and used 2 prior terrestrial models that have been calibrated with country-specific data (in contrast to the Steinkamp study that used one model that was not calibrated for New Zealand). It is worthwhile that these improvements have been made to confirm the general conclusion of the Steinkamp study that the observations show strong $CO_2$ uptake by forests in New Zealand.

The paper is very well written and worthy of publication. My main comment is that it would be worthwhile putting the key details of the inversion methodology into the main part of the paper (e.g. move Section S1 to the main paper, or if condensed then include that the inversion is solved analytically, for fluxes in 25 regions at weekly resolution, there is a smoothing on week-to-week variations). The paper currently doesn't really describe well enough what are the unknowns that are solved for - are these scale factors for the regional fluxes for 25 regions x number of weeks? And also that the spatial flux pattern within regions is maintained. As part of this, I would recommend moving Fig S1 from the Supplement to the main part of the paper, which will also help with some of the discussion later. Even though all of the details are in Steinkamp, I believe the main details of the inversion should be repeated here in the main paper, as the inversion is so central to the study.

We thank Reviewer #1 for their careful revision of our paper and feedback. As suggested by both Reviewer #1 and #2 we have moved the inversion description from the Supplementary Information (S1) to the main paper (now Section 2.1) and provided additional information about the inversion system, outlined below.

Specific comments:

Line 139 - I would emphasise that the custom NZCSM was at ~1.5 km (i.e. "... input data for the period 2014-mid 2016 at 1.5 km (herinafter ..."). Also, at line 146, I would put "... so the mid-term switch from NZLAM at ~1.5 km to NZCSM-generated input data at ~12 km spatial resolution was considered ...". Neither of these changes are critical, but I think they would aid clarity.

We have added the information that the custom NZCSM model was also at a ~1.5 km spatial resolution (Line 171) and specified the spatial resolution between NZCSM and NZLAM when discussing the switch between models (Line 178).

Line 153 - Perhaps change to "time-disaggregated modelled footprints ..." or add other information to explain the disaggregated footprints a bit more.

We have modified the text to say time-disaggregated instead of disaggregated for clarity (Line 185).

Line 188 - there's not much Northwest at Lauder with NZLAM, mainly north.

We have corrected the text to North only (Line 221).

Fig 5 - I would put the NZLAM to the left of NZCSM, in the order they are used in time, so either (Meas, NZLAM, NZCSM) or (NZLAM, NZCSM, Meas) - the 2nd option is probably better. In the caption, I would put '(top)' after 'Baring Head', and '(bottom)' after 'Lauder'.

We thank the reviewer for the suggestion. We have corrected the plot and figure caption as suggested, the updated order is: NZLAM, NZCSM, Meas (Page 11).

Line 197 - I presume this sentence means that the prior fluxes were used "with the observations in the inversion" to estimate the posterior fluxes. It could be misunderstood how it is.

That is correct. The prior fluxes were used with observations (and a transport model) to estimate the posterior fluxes. We have updated the sentence for clarity (Line 231).

Line 312 - What does it mean that you "used" the individual flux components? These components were both estimated in the inversion?

While Gross Primary Production (GPP) and Ecosystem Respiration (ER) were not estimated in the inversion, our inversion system provides an estimate of the Net Ecosystem Exchange (NEE). However, we used the individual GPP and ER components to define the uncertainties for the prior NEE fluxes. We used the individual terms, instead of only NEE, to avoid low uncertainties at times, especially in spring and autumn, when fluxes were very small and could switch between negative and positive. It also provided a better representation of the $CO_2$ seasonal cycle in the uncertainty term (i.e., leading to lower uncertainties in winter when both GPP and ER were small).

Line 143 - It would help with this text if Fig S1 was in the main paper. In the previous sentence, the region name is mentioned before the number, I like this better than mentioning the number first (you do need to mention both).

We have moved Figure S1 into the main paper (now Figure 1, Page 5), as suggested.

Figs 10 and 11 (and perhaps elsewhere) - The captions say 'air-land flux', but a negative value in the figures indicates a positive air-land flux (uptake) and a positive value in the figures indicates negative air-land flux (source, or land-to-air flux). This could be misleading, and worth specifying in the caption what positive and negative values indicate, and possibly using a different term from air-land flux to avoid the implication of a direction of flow for a positive value. Steinkamp's Fig 5 y-axis label is land-to-air flux.

We have modified the 'air-land' (and air-sea) flux term to 'land-to-air' (and sea-to-air) flux in the figure caption and throughout the text (both main text and Supplement) and stated that negative values suggest an uptake while positive values suggest a net source of $CO_2$ (Line 110).

Line 510 - "the overestimated sink is less pronounced" - this could be expressed more clearly.

We have added additional text to clarify (Line 547). Specifically: *"for later years the overestimated sink due to the diurnal cycle bias is less pronounced, and the results even suggest and underestimated sink (Fig. S16)."*

Fig 17 - Put a thin black zero line over the top of the plots so it is clear how the data compares to zero. Also, adding a scale and tickmarks to the right side of the right plots would help too.

We have added the zero line to each subplot (red colour, since black was not visible enough), and tickmarks on the right side (Page 28).

Line 556 What two features does "Both features" refer to? The previous sentence only mentions one feature (supressed autumn/winter respiration).

We have corrected the text (Line 597), we were only referring to the supressed autumn/winter respiration, hence this was a mistake in the text.

Line 572 - "in the prior" - is this supposed to be "compared to the prior"? Or "in the prior" if it was included in the prior model? This part of the sentence is not clear.

It is "in the prior ", referring to the exclusion of the animal respiration process in the prior model (Biome-BGCMuSo). We have updated the text to clarify (Line 613).

Line 751 - 'less sensitive' that what? Than to the other regions?

That is correct, we were referring to the other regions. We have updated the text to clarify this statement (Line 793).

Section S1 - What are the unknowns that are solved for in the inversion? I.e. what is x? Are they scale factors for the priors in the 25 regions? As mentioned above, some details of the inversion system should be described in the main paper, and I would put all of Section S1 into the main paper, with a bit of extra detail.

We have moved the entire S1 Section into the main paper (Section 2.1), and we have provided additional information about the inversion system. Specifically, '*Our inversion system was estimating absolute net $CO_2$ fluxes, rather than scaling factors, for 25 geographic regions (Fig. 1) on a weekly scale, with negative land-to-air fluxes suggesting a net $CO_2$ sink and positive values pointing to a net source. Since we estimated regional fluxes, the spatial flux pattern within regions was maintained.*' (Line 109).

Typographical errors/technical corrections:

Line 57 - It could be useful to add '(boundary conditions)' after 'background values', as some other studies use this terminology. Fixed/Added (Line 57)

Line 42 "scalMOLes" Fixed (Line 42)

Line 173 - replace '.' with 'see', Replaced (Line 206)

Line 229 - Fore New Zealand Fixed (Line 265)

Line 324 - move comma "Australian, region" Fixed (Line 358)

Line 340 - add "such" - "such as the" Added (Line 374)

Line 403 - 'report' -> 'reported. Give a reference for the Inventory here. Fixed and reference added (Line 440)

Line 479 - 'process' -> 'processes' Fixed (Line 516)

Line 543 - 'owing' -> 'showing' Fixed (Line 583)

Line 643 - add 'on the' after 'depending' Added (Line 684)

Line 771 - Check the end of the sentence 'and other .' Checked and fixed to 'and other regions' (Line 814)

**Reviewer 2**

We thank Reviewer #2 for their comments and suggestions on our manuscript "Inverse modelling of New Zealand's carbon dioxide balance estimates a larger than expected carbon sink". In the text below, we have included all the original reviewer comments and suggestions in black, followed by our response and relevant manuscript changes in red. Page and line numbers refer to marked-up manuscript version.

This reviewer feels this paper is in scope for ACP, presenting atmospheric top-down carbon dioxide estimates across New Zealand, using two in-situ sites (North and South island) across a 10-year period. The key result for the paper is a larger carbon sink than previously reported compared with both bottom-up estimates and from the study this paper builds on. The potential sources for this difference are presented and discussed. This also presents an updated biospheric flux model tailored to the region.

This paper is thorough and well written, includes interesting and well-justified conclusions, has appropriate references to current literature throughout and some novel work which could prove useful for the community. The scientific methods are well discussed, with details given on the sets of inputs to the inversion model. Some details for the inversion model itself are included in the Supplementary Information but as this is the focus for the paper these methods should be included and discussed in the main text (as also noted and expanded upon by Reviewer 1).

We thank Reviewer #2 for their careful revision of our paper and feedback. As suggested by both Reviewer #1 and #2 we have moved the inversion description from the Supplementary Information (S1) to the main paper (now Section 2.1)

Specific comments below:

L153: "Our inversion used 4-day integrated air concentration (i.e., footprints, units g s m-3, Fig. 3), averaged throughout the Planetary Boundary Layer (PBL)". What is meant by averaged throughout the Planetary Boundary Layer (PBL) here? Was this taken at various heights above the surface based on the PBL values in NAME meaning this was variable for each footprint?

That is correct, each footprint (i.e., the vertical extent of the footprints) will vary with each grid box and time step depending on the height of the modelled PBL. The boundary layer height field is available to NAME III on model levels (70 levels ranging from the surface to 40km above the surface). Any averaging is done from the surface up to the diagnosed boundary layer height over the requisite number of levels. A detailed description is available in Section 1.4.1.1 and Table 1.18 in https://code.metoffice.gov.uk/doc/name/vn7.2/docs/namedoc_A01.pdf (accessing the document requires registration with MetOffice).

L159: "By the end of the 4 days, most of the particles had left the model domain." Has this been tested for the footprints involved in this study as well as Steinkamp et al., 2017? A 4-day period seems short but is this due to small domain and large wind speeds?

This has been tested both in Steinkamp et al., (2017) and for this study, although the results are not shown in the paper. NAME III provides a specific diagnostic that tracks how many particles leave the modelled domain. Figure 1 shows an example of the analysis for year 2014 for both Baring Head and Lauder.

[Figure]

Figure 1. Mean and standard deviation of particle counts over time for Baring Head (top) and Lauder (bottom) from backward trajectory simulations over 96 hours (~4 days). The solid lines show the mean number of particles remaining, while the shaded regions represent ±1 standard deviation. Particles are released at hour 0, with numbers decreasing as they exit the domain. The dashed line at y = 0 marks complete particle loss.

For each measurement time, we release 10,000 particles (y-axis), and the majority of the particles leave the domain after 4-days (i.e., 96 hours, ~0 particles). Using 2014 as an example, for Baring Head on 80% of the days more than 90% of the particles left the domain, while for Lauder on 85% of the days more than 90% of the particles left the domain. The particles that remain in the domain are during days with very low wind speed. Other years follow the same pattern.

Section 2.2.1 Atmopsheric model transport validation: Why are wind speed and wind direction the most important meteorlogical parameters to investigate here? Were there other parameters which may be significant as well?

We appreciate the reviewer's comment and the opportunity to clarify our choice of meteorological parameters for validation. Wind speed and wind direction are the primary drivers of transport in NAME III, directly influencing the dispersion and advection of atmospheric tracers. Since NAME III is a Lagrangian transport model, accurate representation of these parameters is critical for ensuring that air parcels are correctly advected from source regions to measurement sites. In addition to wind fields, we also evaluated planetary boundary layer (PBL) heights at Lauder (Figure S5), recognizing that vertical mixing is another key factor affecting particle transport. However, the validation was limited by the availability of verification data. We believe that wind speed, wind direction, and PBL height represent the most essential diagnostics for evaluating transport accuracy in this context. Furthermore, we provide a comparison of the topographical representation in the model (Figure S4), demonstrating the improvements achieved with higher-resolution meteorological input data (i.e., NZCSM).

Section 2.4 Prior and posterior uncertainties: Would be good to see justification for the uncertainty terms chosen including:

 - "we fixed (i.e. summed) 50% of the uncertainty term"

 - "The uncertainity for pine fluxes from CenW were assumed to be 30%, 30%, 60% of the NEE, GPP and ER flux magnitudes"

 - what was the justification for these choices for uncertainties? Do we know they are representative (or that the extremes included within the sensitvity tests are justified)?

The fixed (i.e., summed) 50% uncertainty term was chosen to ensure realistic posterior uncertainties, as lower values appeared small given known transport and model-data mismatch errors. This adjustment accounts for systematic uncertainties and underrepresented variability, preventing overconfidence in the estimates. Sensitivity tests confirm that our approach captures a reasonable range of uncertainty while avoiding extreme underestimation (Figure 15, Page 23).

In terms of the pine fluxes from CenW, the model had been parameterised against growth data from 1309 observations from 101 stands located throughout New Zealand (Kirschbaum & Watt, 2011). That data set covered stands growing under all the environmental conditions over which *Pinus radiata* is grown in New Zealand. Comparison between observed and modelled volume growth was described with a Nash-Sutcliffe model efficiency of 0.853, hence providing high confidence in the model's performance in describing the overall long-term growth potential (Kirschbaum & Watt, 2011).

Translating that into confidence of short-term NEE fluxes adds some additional temporal and spatial uncertainty, although both were constrained by the broad coverage in the observations. Hence, an NEE uncertainty of 30% seemed conservative given the model efficiency of 0.853 in the underlying model-data comparison. As the major flux driving NEE is GPP, and, given that part of the uncertainty in NEE results from uncertainty in ER, we assigned the same uncertainty to GPP as had been determined for NEE.

For ER there are two additional main sources of uncertainty:

1) The proportion of GPP captured in ultimate growth as reflected in NEE can vary through physiological factors, such as the allocation to different biomass components, such as foliage and branches, that senesce eventually and are lost through respiration, versus allocation to stems that are retained in the measured eventually observed wood growth;

2) The respiratory contribution from the decomposition of organic matter or other plant residues. In this context for second and later-rotation pine stands, the decomposition of dead coarse roots from preceding rotations are particularly important through their quantitatively large contributions. We estimated that approximately ¼ of carbon fixed during the growth of a pine stand is allocated to the root system. Over the next rotation, that carbon will eventually be lost by respiration after the root system is killed during harvesting. Uncertainties relate to the timing of that loss, its magnitude and whether stands are first rotation or subsequent rotation stands.

Given these two factors, we estimated the uncertainty of ER fluxes to be twice as large as the uncertainty of GPP fluxes

References: Kirschbaum, M. U. F., & Watt, M. S. (2011). Use of a process-based model to describe spatial variation in Pinus radiata productivity in New Zealand. Forest Ecology and Management, 262(6), 1008-1019.

Table 2: Exclusion of Test 9 and 10 from the table itself (and details included in the caption) is a little confusing when comparing to Figure 14. Would be good to see the details of these tests included within the table.

Details of both Test 9 and 10 are now included in Table 2 (Page 24).

Supplementary Info

SL54: "NAME III vn6.5 dispersion model for the years 2014-2020 while for the period 2011-2013, we retained the original NAME III vn6.1 simulations" - was any comparison made between the outputs between the two NAME versions? Did this make any difference to the sensitivity profiles?

A comparison between the two NAME III versions was conducted for ~6 months in 2014 (January to July) when compatible input data were available for both models. The results showed minimal differences, with no significant impact on the sensitivity profiles. The core science and transport schemes relevant to the inversion remained unchanged between NAME III vn6.1 and vn6.5. The updates primarily involved technical modifications or changes to schemes that were not utilized in this study, ensuring consistency in our inversion framework.

SL68: "00Z" - does this mean UTC at midnight?

Yes, "00Z" refers to midnight 00:00 UTC and equivalent to 12 UTC in New Zealand, following a convention in Numerical Weather Prediction (NWP) to use UTC or "Z" as a consistent time reference for global coordination.

Figure S6: Consider updating the red-green in this plot to something colour-blind friendly (this could include changing one of the maker styles to be distinctive).

We thank the reviewer for this suggestion, we have updated the colours of the figure (and figure caption), designed to be distinguishable for people with colour vision deficiencies (now Figure S5, Page 7).

Table S3: For the "Area (m2)" column is the area itself known to this level of precision?

We appreciate the reviewer's comment. The high precision in the "Area (m$^2$)" column was a result of the table being generated automatically from the inversion output files, which contained area values with the specified level of precision. We recognize that such precision is unnecessary for reporting purposes and we have now removed the decimal values for clarity (Page 14).

Figure S17: "Diurnal cycle test results for the land inversion regions." Caption is not too descriptive and requires other details to understand the context for this. Please add a few more details here to this caption more standalone.

We have added additional information about the figure. Specifically: *Diurnal cycle test results (Section 4.3, main paper) for the land inversion regions. The figure shows the timeseries of prior (black) and posterior (green) fluxes for each land inversion region (i.e., region 1-15), along with corresponding annual values. The numbers above the annual plots represent the prior (top) and posterior (bottom) annual fluxes in units of Tg CO$_2$ yr-$^1$. Identical prior and posterior fluxes indicate no systematic bias in the inversion system, while any differences suggest a potential over- or underestimation of CO$_2$ sources or sinks.* (now Figure S16, Page 31)

Typos and grammer:

- L42    scalMOLes --> scales Fixed (Line 42)

- L123-124 (Terao et al., 2011; Yamagishi et al., 2012; Müller et al., 2021)) - Bracket within a bracket here, this should be typeset to remove this. Fixed (Line 157)

- L229: Fore --> For Fixed (Line 265)

- L256: Front quote around 'other forests' is a back rather than a front quote Fixed (Line 292)

- L726: "the inverse model detect" --> grammer issue here Fixed (Line 767)

Supplementary Info

- SL68: creata --> create Fixed (Line 72)

- Figure S16: Tabel 3. --> Table Fixed (now Figure S15, Page 31)

- Figure S19: Timseries --> Timeseries Fixed (now Figure S18, Page 33)

**Reviewer 3**

We thank Reviewer #3 for their comments and suggestions on our manuscript "Inverse modelling of New Zealand's carbon dioxide balance estimates a larger than expected carbon sink". In the text below, we have included all the original reviewer comments and suggestions in black, followed by our response and relevant manuscript changes in red. Page and line numbers refer to marked-up manuscript version.

Review of "Inverse modelling of New Zealand's carbon dioxide balance estimates a larger than expected carbon sink" by Beata Bukosa et al.

I read this interesting manuscript submitted for publication in ACP. The paper uses observations of CO2 from two sites in New Zealand, an inverse model that uses for transport modelling and land fluxes from two terrestrial ecosystem models plus oceanic exchange fluxes. The paper is very detailed and well written. However, as it appears from the writing, the authors are finding it difficult to reconcile the inversion estimated flux of CO2 with the country level estimations by bottom-up methods or those predicted by the land models. I have one concern that is the handling of the boundary condition which is most tricky in original model when simulating the long-lived atmospheric constituents. I am fairly convinced that this paper should be published, but a follow-up study with a variety of boundary conditions would be of interests for the research community.

We thank Reviewer #3 for their careful revision of our paper and feedback. We have provided additional information about the boundary conditions in our answers below.

Specific comments:

Line 42: typo – scalMOLes Fixed (Line 42)

Line 51-53: a bit of overstatement here, given that you are still struggling with reconcillation of bottom up and top down estimations. Of course, it is possible to get good comparison for a specific set of results which may not be universally applicable.

We appreciate the reviewer's comment and understand the concern regarding the reconciliation of bottom-up and top-down $CO_2$ flux estimates. While differences between these approaches are expected, especially given the complexities of $CO_2$ flux estimation, our intent was to highlight that national-scale inverse modelling remains the most successful and widely adopted approach for independent verification of national $CO_2$ budgets. The fact that only a few countries have successfully implemented these methods reinforces the challenge of applying them to $CO_2$, rather than diminishing their validity. That said, we recognize the importance of clarity and would be happy to adjust the wording if the reviewer finds it necessary.

Figure 2: this is one of the most important plots in my view because the clear offset with background, as seen from panel b at Lauder, will produce more pronounced sinks around Lauder, that is, the South Island of New Zealand.

That is correct, the measurements itself are already highlighting a net $CO_2$ sink in the South Island (now Figure 3).

Line 121-124: i think this is likely a wrong construct of the inversion, when the zonal winds are very strong over NZ! You probably need to use a global model providing 3D concentrations of CO2 for use as a background, possibly after adjusting to BHD & TF5 measurements.

We agree that the South and Northern baseline can introduce uncertainties in the background due to zonal winds. To evaluate the sensitivity of our results to this approximation, we have also tested a 3D concentration model (CarbonTracker, Section 4.2) as a background. Implementing CarbonTracker led to some differences in the national $CO_2$ sink (Figure 15). However, the differences were within uncertainties and they did not impact our results and conclusions in the

paper. We have decided to keep the BHD & TF5 measurements as the background in our base inversions, since the 3D field are also subject to uncertainties.

Line 144: how do you define the planetary boundary layer-does it change with hour of the day? What data and method is used to determine the PBL height?

The planetary boundary layer height is a diagnosed quantity from the model. It is dependent on the boundary layer type diagnosed by the model's boundary layer scheme on every model (dynamic) timestep in each grid box. Therefore it can change throughout the day and we output it every 30 minues from the New Zealand Convective Scale Model (NZCSM). A detailed description is available in Section 3 of https://code.metoffice.gov.uk/doc/um/latest/papers/umdp_024.pdf (accessing the document requires registration with MetOffice).

Figure 8: as expected from Figure 2b, the inversion results for South Island, in particular south of Lauder, show strong negative values or CO2 sink.

That is correct (now Figure 3b).

Figure 9: the issue of strong sink south of Lauder is further clear from this plots. The green bars are bigger compare to the black ones for the regions 13 to 15.

That is correct (now Figure 10).

Figure 17: are these a priori or a poste? in any case they can clearly see the observed concentrations were lower than the model in the case of Lauder . And then as expected you would need more sink in the south ern part of the South Island (Fig. 9)

These are the posterior values (minus the observations), which is specified on Line 558: '*Residuals represent the differences between the modelled and measured $CO_2$ mole fractions, with the modelled values being the optimized $CO_2$ mole fractions by propagating the posterior flux estimates through the inversion.*'

Figure 17: caption can start with "Model – measured …"

We have modified the caption (now Figure 18).